# Phenotypic memory in *Bacillus subtilis* links dormancy entry and exit by a spore quantity-quality tradeoff

Alper Mutlu[1,2,3], Stephanie Trauth[1,2,3], Marika Ziesack[1,2], Katja Nagler[1,3], Jan-Philip Bergeest[1,4,5], Karl Rohr[1,4,5], Nils Becker[1,6], Thomas Höfer[1,6] & Ilka B. Bischofs[1,2,3]

Some bacteria, such as *Bacillus subtilis*, withstand starvation by forming dormant spores that revive when nutrients become available. Although sporulation and spore revival jointly determine survival in fluctuating environments, the relationship between them has been unclear. Here we show that these two processes are linked by a phenotypic "memory" that arises from a carry-over of molecules from the vegetative cell into the spore. By imaging life histories of individual *B. subtilis* cells using fluorescent reporters, we demonstrate that sporulation timing controls nutrient-induced spore revival. Alanine dehydrogenase contributes to spore memory and controls alanine-induced outgrowth, thereby coupling a spore's revival capacity to the gene expression and growth history of its progenitors. A theoretical analysis, and experiments with signaling mutants exhibiting altered sporulation timing, support the hypothesis that such an intrinsically generated memory leads to a tradeoff between spore quantity and spore quality, which could drive the emergence of complex microbial traits.

[1] BioQuant Center of the University of Heidelberg, 69120 Heidelberg, Germany. [2] Center for Molecular Biology (ZMBH), University of Heidelberg, 69120 Heidelberg, Germany. [3] Max-Planck-Institute for Terrestrial Microbiology, 35043 Marburg, Germany. [4] Institute of Pharmacy and Molecular Biotechnology (IPMB), 69120 Heidelberg, Germany. [5] Department of Bioinformatics and Functional Genomics, German Cancer Research Center (DKFZ), 69120 Heidelberg, Germany. [6] Division of Theoretical Systems Biology, German Cancer Research Center (DKFZ), 69120 Heidelberg, Germany. Correspondence and requests for materials should be addressed to I.B.B. (email: ilka.bischofs@mpi-marburg.mpg.de)

In response to fluctuations in nutrient availability, bacteria switch between a proliferative vegetative state and a dormant state. One striking example of this capacity is the ability of many Gram-positive bacteria to differentiate into endospores and to revert from the dormant state through spore germination and subsequent outgrowth, thereby resuming the vegetative growth mode[1]. The molecular processes underlying these transitions have been particularly well characterized in the model organism *Bacillus subtilis*. Both vegetative cells and spores possess a signal transduction system that channels information relating to external (and internal) conditions[2,3] and enables each to decide whether to trigger a cascade of molecular events that transforms one cell type into the other[4,5]. Traditionally, sporulation and spore revival control have been studied separately, although both processes jointly determine survival in fluctuating environments over longer timescales.

A striking degree of phenotypic variability is common to both sporulation and spore revival, correlating with variable signaling activities. In *B. subtilis*, the coexistence of vegetative and sporulating cells following a nutrient downshift is the result of "heterochronic," i.e., temporally variable, activation of the sporulation master regulator Spo0A via signaling through the sporulation phosphorelay[6–10], whereas variability in nutrient-induced spore revival is at least partially controlled by variability in the levels of the spores' germinant receptors[11,12]. In each case, cell-to-cell heterogeneity in the propensity to undergo conversions from one cell type to the other has been proposed to facilitate adaptive bet-hedging[13–17]. Thus, given that the two transitions are regulated by separate pathways and that each transition might have its own specific needs if it is to support the survival of the species, there might be little a priori reason to expect that variability in one process should be correlated with variability in the other.

On the other hand, several population-level studies have found that environmental conditions, during[18,19] and even after sporulation[20–22], modulate the average resistance of spores, as well as their capacity for revival, by influencing their molecular composition. Furthermore, a considerable portion of the spore's proteinaceous cargo derives from its progenitor cell[23,24]—presumably through carry-over of stable proteins[25]—although little is known about their functional relevance for spore revival. One may thus consider the history-dependent portion of a spore's composition as a "memory" of the past that could influence a spore's future revival response. Indeed, memory effects may couple the heterogeneous dynamics of sporulation and spore revival at the level of the life histories of individual cells. However, experimental evidence for this hypothesis has been lacking and the potential consequences of such memory for linking of sporulation to spore revival are unclear.

Here, we address these questions by imaging life histories of individual *B. subtilis* cells under changing nutrient conditions. We find that heterochronic sporulation results in the formation of spore subpopulations with a long-term memory that controls their responses to nutrient-induced spore revival. This memory stems from the growth and gene-expression history of individual cells prior to the onset of sporulation, and we identify alanine dehydrogenase (Ald) as an important molecular determinant of this memory. High levels of Ald in the spore core support outgrowth following alanine-dependent germination, provided sufficient alanine is in the medium. Moreover, this intrinsically generated phenotypic memory leads to the emergence of a tradeoff between spore quantity and spore quality.

## Results

### Correlative imaging of sporulation and spore revival.
Long-term single-cell time-lapse microscopy using gel pads is a

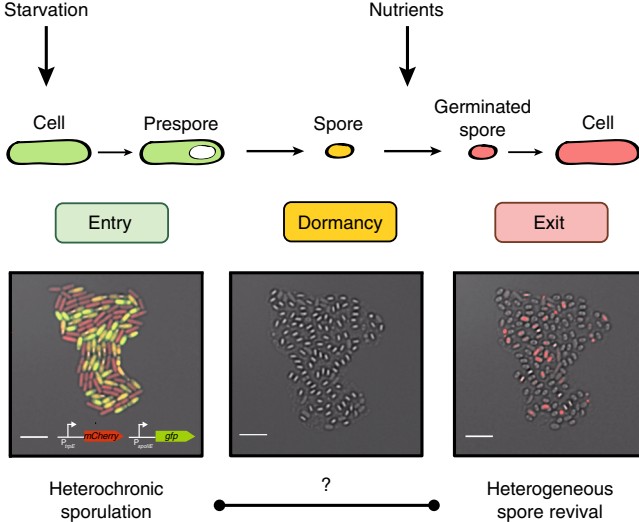

**Fig. 1** Tracking the life histories of individual bacteria to study the role of memory in the *B. subtilis* life cycle. Left: image of a heterogeneously sporulating microcolony. Sporulating cells express GFP from the $P_{spoIIE}$ promoter that is activated at the onset of sporulation. Other cells delay sporulation. They express mCherry from the $P_{trpE*}$ promoter that is active during vegetative growth. Center: image of the same microcolony after sporulation is complete. Right: in response to nutrients, most spores germinate and some grow out. Scale bars: 5 μm. Strain: BIB1019

powerful method for examining dynamic processes in bacteria at the cellular and molecular levels[26–28]. However, limitations on our ability to perturb and control bacteria by changing cell conditions have so far been a serious drawback, and the analysis of                                    the *B. subtilis* life cycle has focused on either sporulation[6–10] or spore revival[29–31]. We first developed an assay to track the life histories of individual bacteria under changing nutrient conditions (Fig. 1). The use of a motorized microtiter-plate stage allowed us to repeatedly mount and retrieve the plate and reliably relocate a given position with approximately micrometer accuracy. This enabled us to alter the environmental conditions for cells by adding nutrients or other factors to the gel pads, and then resume imaging to follow the response of precisely the same cells within minutes. In brief, cells that had been grown in rich casein hydrolysate (CH) media were subjected to a nutrient downshift, spotted on agarose pads containing sporulation medium (SM), and imaged until the entire population had completed sporulation (~4 days). We then applied a nutrient stimulus and monitored the subsequent upshift response of the spores. In this way, we were able to directly investigate the correlations between sporulation and the subsequent behavior of the resulting spores at the single-cell level. Spore revival is traditionally separated into two phases[32]: germination (in which germinants trigger the spore to rehydrate and dismantle its protective structures) and outgrowth (in which the spore reactivates macromolecular synthesis and resumes vegetative growth). Both are complex multistage processes, which rely on a sequence of molecular events that gradually, but irreversibly, turn the dormant spore into a vegetative cell[4,32,33]. We first searched for evidence that history affects the overall completion of spore revival. For a subset of experiments, we subsequently increased the temporal resolution to study effects on germination.

**Spore revival correlates with the timing of spore formation.** For each individual spore, we recorded the time at which differentiation into the dormant state occurred, as indicated by

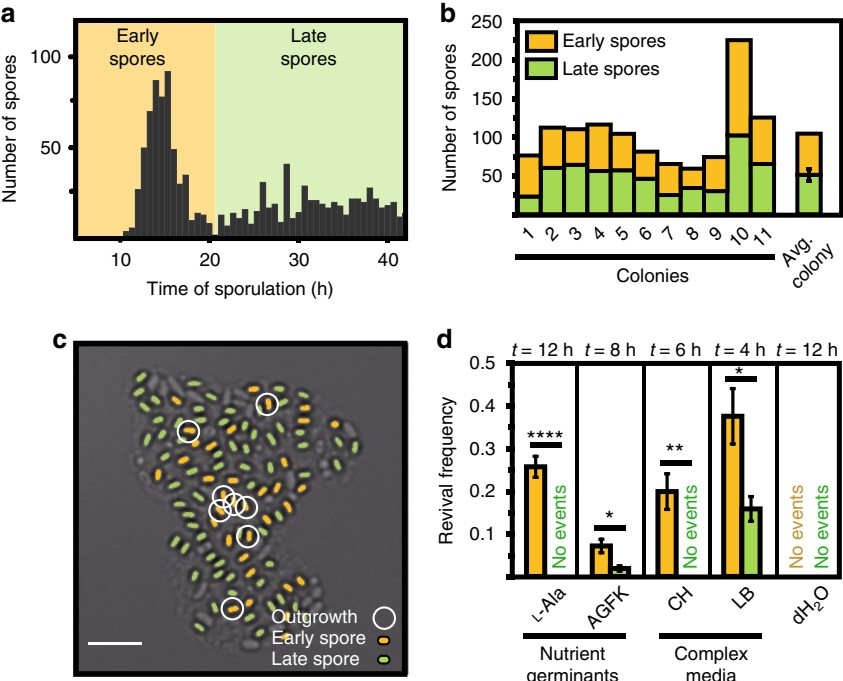

**Fig. 2** Nutrient-induced spore revival correlates with the timing of sporulation during starvation. **a** Distribution of sporulation times. Spores were classified into early (yellow) and late spores (green). The number of tracked spores is $n_s = 1013$ from $n_c = 10$ microcolonies obtained from $n_e = 3$ independent experiments. **b** Total number of early and late spores in different microcolonies. **c** False-colored bright-field image of a microcolony generated by a single vegetative cell, visualized after 90 h of starvation, and showing the distribution of mature spores. Yellow denotes early spores and green late spores. The subset of spores that grew out in response to stimulation with L-alanine within the first 12 h is circled. Scale bar: 5 µm. **d** Revival frequencies of early and late spores in response to stimulation with L-Ala (L-alanine), AGFK (L-asparagine, glucose, fructose, potassium), CH (casein hydrolysate medium), LB (luria broth), and H₂O (water). See Methods for details. Strain: BIB1019, data: mean ± SEM from $n_c \geq 4$ ($n_s \geq 500$), unpaired $t$ test: *$P \leq 0.05$, **$P \leq 0.01$, ***$P \leq 0.001$, ****$P \leq 0.0001$

the first appearance of a bright pre-spore (Fig. 2a). In accordance with previous findings[6–10], we observed a substantial amount of intercellular variability in the onset of sporulation. Following the nutrient downshift, cells at first continued to divide and grew into small microcolonies. Subsequently, a subset of cells in each microcolony initiated sporulation. This resulted in a first wave of pre-spores, which became visible between 10 and 20 h after the nutrient downshift. We refer to these spores as "early spores". Other cells in the developing microcolonies delayed sporulation and continued to divide at a slow rate for many more hours. Thus, a spore was classified as a "late spore" if the pre-spore had formed after the first wave of sporulation events, typically 20 h or more following the nutrient downshift. While the total number of spores varied from one microcolony to another, on average about one-half of all resulting spores fell into the early and late spore classes, respectively (Fig. 2b) and there was no obvious pattern in their spatial distributions (Fig. 2c).

After 4 days of starvation, both early and late spores had been released from the sporangia, which was taken as an indication that development was complete[1,5]. We then applied a nutrient upshift by adding L-alanine. When added to pure buffer on its own, L-alanine is known to trigger germination by activating specific germinant receptors[34,35]; when applied to exhausted SM, L-alanine can in addition support spore outgrowth[36]. Under our conditions, following stimulation with L-alanine, most spores germinated (>90%). However, only a subset of spores in each microcolony had grown out to complete spore revival, by the end of our 12-h observation period. Notably, outgrowing spores were found both at the center and the periphery of spore microcolonies (Fig. 2c). Strikingly, when we correlated outgrowth with spore differentiation times, we found that all outgrowing spores had

initiated sporulation early (Supplementary Movie 1). We then used different nutrient germinant solutions (that act via distinct sets of germinant receptors[37,38]) or more complex growth media and compared the revival responses of early and late spores by quantifying the revival frequency $f_r$, defined as the fraction of spores that grew out in a specified time interval in each case. Under all tested conditions, early spores had a higher revival frequency than late spores (Fig. 2d).

**Early spores germinate faster and use alanine for outgrowth.**
To resolve the relative contributions of germination and outgrowth phases to the differential revival success of early and late spores, we stimulated spores with different concentrations of L-alanine. We determined the germination frequency $f_g$ by measuring the fraction of spores that have lost their refractivity due to germination (Fig. 3a). Notably, the germination frequency in each subpopulation decreased in very similar ways for early and late spores as the alanine concentration was reduced (Supplementary Fig. 1). However, early spores germinated about twice as fast as the late spores. This effect was observed upon application of either a saturating stimulus resulting in germination levels >90% after 2 h (Fig. 3b) or a 20-fold weaker stimulus, by which only ~40% of all spores germinated (Fig. 3c). In contrast to the strong stimulus that allowed some of the germinated (early) spores to grow out as shown before, the spores stalled in the germinated state when the concentration of alanine was too low. When we subsequently added a second stronger alanine stimulus to these spores (Fig. 4a), we observed outgrowth, but again, only among early spores (Fig. 4b). Specifically, none of the pre-germinated late spores grew out, whereas a substantial fraction of the pre-

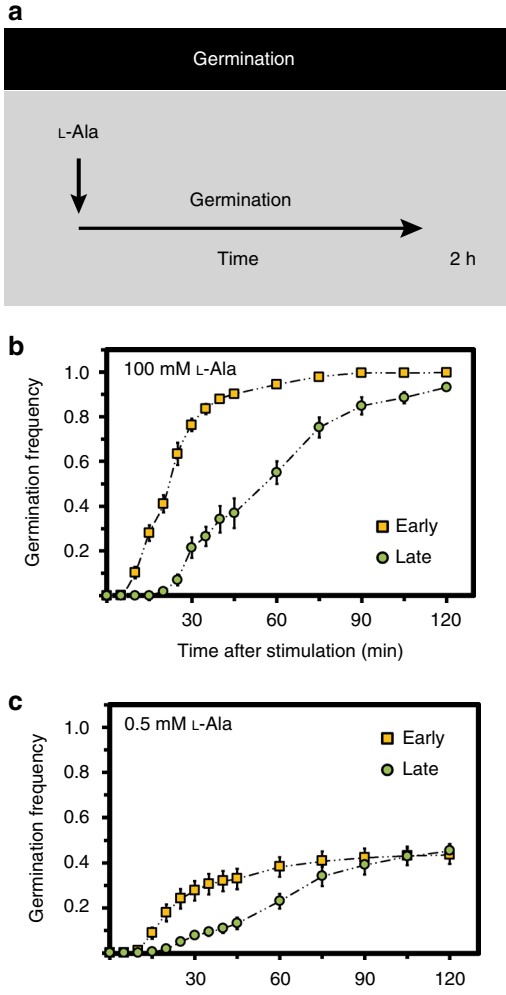

**Fig. 3** Early spores germinate faster than late spore in response to the nutrient germinant L-alanine. **a** Germination dynamics was assessed by measuring the fraction of spores that lost refractivity as a function of time in response to the nutrient germinant L-alanine. See Methods for details. **b** Germination frequency of early and late spores in response to a strong stimulus and **c** a weak stimulus. Strain: BIB1019, data: mean ± SEM, $n_c = 9$ ($n_s > 400$)

germinated early spores responded. Thus, although early and late spores had the same germination frequency (Fig. 4c), their outgrowth frequencies $f_o$, defined as the number of outgrowing spores divided by the number of pre-germinated spores, were distinct (Fig. 4d). These experiments thus indicated that L-alanine is a germination trigger for both early and late spores, but only early spores are capable of further utilizing alanine efficiently to support outgrowth.

**A fluorescent marker can distinguish early from late spores**. Taken together, our time-lapse experiments revealed that, under conditions where spore revival among genetically identical spores is highly variable, successful outgrowth is not determined by chance alone. Instead there exists a striking correlation between the timing of sporulation and the propensity of the resulting spore for nutrient-induced exit from dormancy. Specifically, the early spores germinate faster and are more likely to grow out. This may indicate that sporulation timing indeed influences the properties of a spore. However, although we could not detect any

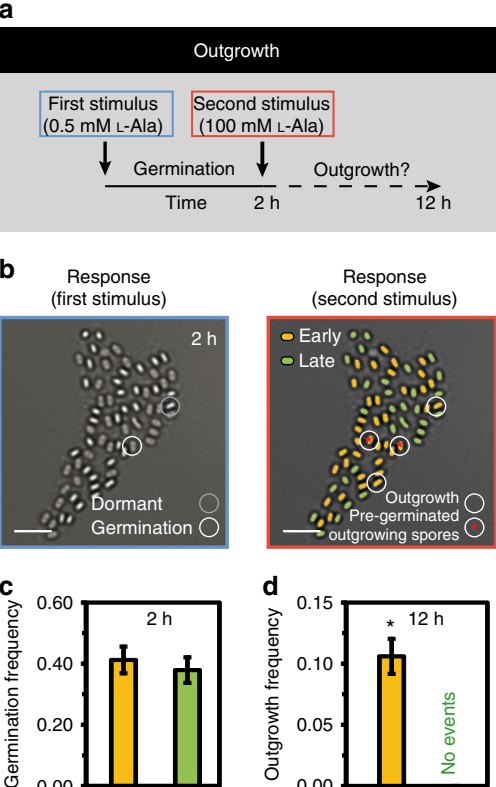

**Fig. 4** Early spores are capable of L-alanine-induced outgrowth. **a** Schematic depiction of the two-step stimulation experiment. Spores stall in the germinated state after the first weak stimulus. In response to the second stronger stimulus, a fraction of germinated spores will grow out. **b** Image of a representative microcolony that has completed the germination response to the first weak stimulus and a corresponding false-colored image that indicates the outgrowth response of all spores to the second, stronger stimulus. Notably none of the germinated late spores grew out. Scale bar: 5 μm. **c** Germination frequencies in response to the first stimulus. **d** Outgrowth frequencies in response to the second stimulus. Strain: BIB1019, data: mean ± SEM, $n_c = 5$ ($n_s > 200$), unpaired $t$ test: *$P \leq 0.05$

obvious spatial pattern in the distributions of early and late spores within microcolonies, subtle microenvironmental effects might nevertheless exist within populations[39]. Additionally, it has recently been shown that *B. subtilis* spores undergo external (e.g., in the spore coat[21]) and internal changes (e.g., in RNA content[20,22] or metabolite levels[40]) over the course of a few days, which have been shown to affect their ability to cope with environmental stress[21] and revive[20,41]. Hence, the late spores may not have had sufficient time to fully mature in our time-lapse experiment.

To facilitate further analysis, we developed a fluorescent marker that is capable of distinguishing between early and late spores. Under sporulation conditions, expression of the signaling protein RapA, which represses entry into sporulation[42], occurs heterogeneously throughout the cell population[43]. In starving microcolonies, cells that transcribe very little *rapA* sporulate early, whereas in cells that activate *rapA* strongly sporulation is delayed[7]. Therefore, a construct containing a *rapA* promoter fused to a gene coding for a stable fluorescent protein should act as an endogenous reporter for the timing of spore differentiation (Fig. 5a). A *rapA* promoter fusion to *mCherry* indeed allowed us to distinguish between the two kinds of spores (Fig. 5b). The distribution of fluorescence levels measured in the spores was

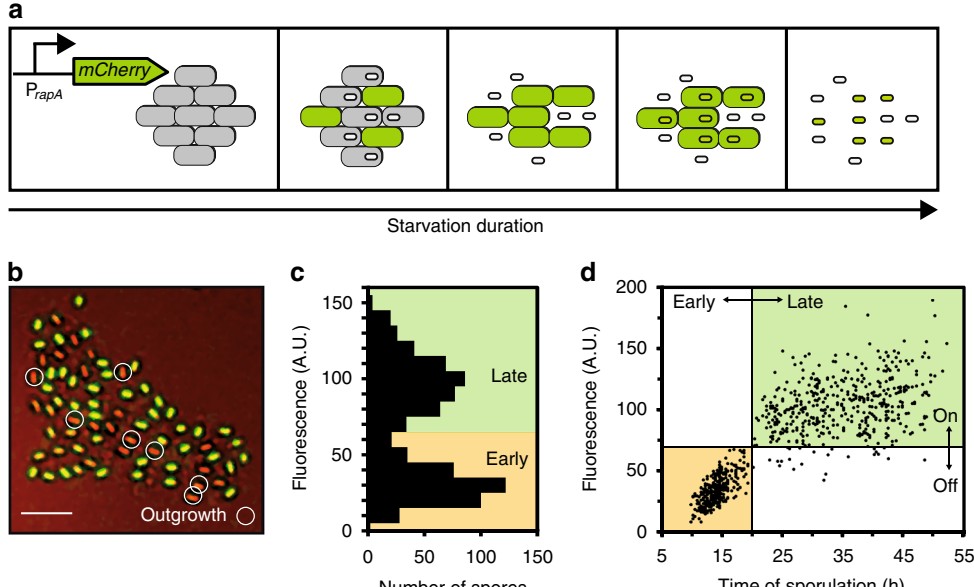

**Fig. 5** A fluorescent marker distinguishes early from late spores. **a** Illustration of the function of the $P_{rapA}$-*mCherry* reporter. **b** Image of a microcolony of spores carrying the $P_{rapA}$-*mCherry* reporter system (BIB1126). The bright-field image (red) was overlaid with the fluorescence image (green). Two subpopulations of spores can be distinguished. The circled spores grew out in response to L-alanine. Scale bar, 5 μm. **c** Histogram of fluorescence intensities of spores ($n_s = 810$). **d** Corresponding fluorescence intensities of individual spores plotted as a function of their sporulation time ($n_s = 810$). Yellow/green areas denote the classification of spores based on sporulation timing

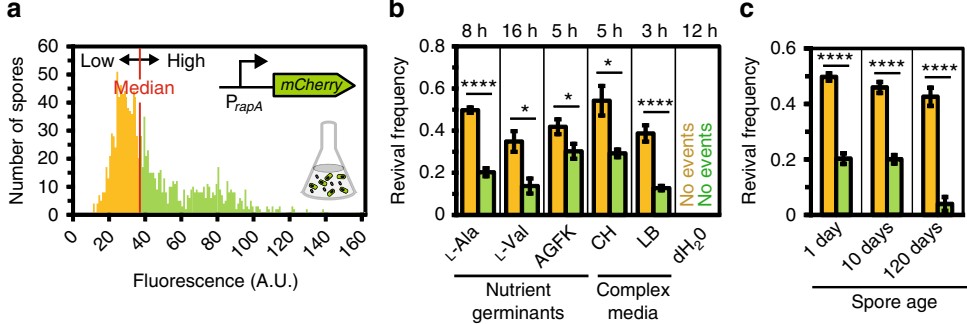

**Fig. 6** Sporulation timing controls nutrient-induced spore revival. **a** Histogram of fluorescence values obtained from spores that carry the $P_{rapA}$-*mCherry* reporter (BIB1126) and were generated in a shaken SM culture. The median is marked in red and was used to group early and late spores. **b** Revival frequencies of early and late spores in response to the indicated nutrients measured at the designated times. **c** Revival frequencies of early and late spores that were kept in buffered saline for different times and then stimulated with L-alanine. Day 1 denotes the day of spore collection. Strain: BIB1126. Data: mean ± SEM, $n \geq 5$ movies ($n_s \geq 900$), unpaired $t$ test: *$P \leq 0.05$, **$P \leq 0.01$, ***$P \leq 0.001$, ****$P \leq 0.0001$

bimodal (Fig. 5c) and the two peaks corresponded well to the ad hoc classification scheme for early and late spores introduced above (Fig. 5d). Moreover, when the spores were induced with L-alanine, only weakly fluorescent spores grew out, as expected (Supplementary Movie 2).

**The history of spore formation controls spore revival.** When spores carrying the reporter were generated in the well-mixed environment of a shake-flask culture using the standard Sterlini–Mandelstam protocol[44], weakly fluorescent spores were produced before more highly fluorescent spores (Supplementary Fig. 2) and the resulting fluorescence distribution was broadly heterogeneous (albeit not distinctly bimodal) (Fig. 6a).We then analyzed spore revival by adding nutrient germinants and more complex growth medium to spores on SM agarose pads (Supplementary Movie 3). For all tested conditions, weakly fluorescent

early spores had a higher revival frequency than highly fluorescent late ones (Fig. 6b). We then increased the interval between the collecting of spores and the nutrient upshift. Collected spores were kept in buffer at room temperature for up to 4 months and then induced with L-alanine. Late spores did not "catch up" with early spores during this time, consistently yielding lower revival frequencies (Fig. 6c). Interestingly, the revival frequencies for both populations appeared to decrease very slowly over time. This might be caused by a loss of responsive spores, e.g., due to spontaneous germination of spores[16] or spore "aging" effects resulting from the degradation of biomolecules[20,40]. In any case, these effects did not equalize the differences between early and late spores. Moreover, the two spore populations also retained their distinct propensities for outgrowth after heat treatment (Supplementary Fig. 3). We thus concluded that the differences between early and late spores must be encoded before or during spore development and that they persist over time.

**Alanine dehydrogenase controls alanine-induced outgrowth**. It is an intriguing hypothesis that cells that delay sporulation might experience altered environmental conditions at the onset of sporulation[18]. Physiochemical constraints resulting from prolonged starvation could affect the development of late spores and render them "defective," e.g., impair their ability to utilize alanine for outgrowth. To investigate this possibility, we focused on alanine dehydrogenase (Ald), a metabolic enzyme that converts alanine to pyruvate and which can influence spore development[36,45,46]. In a co-culture, an *ald* knockout strain sporulated as efficiently as the wild type (WT) and the resulting spores also germinated efficiently when induced with L-alanine. However, in contrast to the WT, none of the *ald* spores grew out (Supplementary Movie 4, Supplementary Fig. 4). Hence, lack of Ald may limit alanine-induced outgrowth. If this were the case, one might be able to reprogram late spores to become capable of outgrowth by supplying Ald. To test this, we engineered the ability to express Ald from an IPTG (isopropyl-β-D-thiogalactopyranosid)-inducible promoter into our fluorescent marker strain. Without induction, the late spores were unable to grow out in response to L-alanine as in the WT strain. We then added IPTG to the pads after the early spores had initiated cell differentiation. Strikingly, the late Ald-induced spores were indeed able to grow out (Supplementary Movie 5). In fact, the revival frequency of late spores was significantly higher than that of the early spores (Supplementary Fig. 5). Notably, expression of Ald facilitated spore outgrowth—but it did not alter germination frequency (Supplementary Fig. 6). Furthermore, Ald expression

did not significantly alter the revival response to AGFK (Supplementary Fig. 7). This indicates that Ald specifically controls alanine-induced outgrowth under our experimental conditions.

**Alanine dehydrogenase contributes to a spore's memory**. To understand how Ald controls the behavior of early and late spores, we analyzed gene expression from the *ald* promoter in single cells with a GFP promoter fusion. Fluorescence first increased and then began to decay approximately at the same time in *all* cells before the onset of heterochronic sporulation (Supplementary Fig. 8). This indicated that gene expression from the *ald* promoter starts from the "on"-state and is subsequently turned down, presumably as alanine levels are depleted from the medium[47]. If transcription of *ald* is downregulated before the onset of sporulation, spores may obtain Ald from their progenitors. This could give rise to a simple mechanism by which information on sporulation timing could be stored in spores. If the production of a factor that limits spore revival ceases during starvation, its level will decay for as long as cells delay sporulation, due to dilution by growth and possibly degradation; thus, less of the factor can ultimately be packed into the spore. This simple dilution-fixation mechanism could result in a stable intrinsic spore memory (Fig. 7).

We thus studied the behavior of a strain that expresses an Ald-mCherry fusion protein from the *ald* promoter (Supplementary Movie 6). Indeed, cells that committed to sporulation early had higher fluorescence levels at the onset of sporulation than cells that continued to divide and gave rise to late spores (Fig. 8a). Moreover, the fluorescence was carried over into the dormant spore (Fig. 8b). As expected, the early spores had a higher fluorescence than the late spores (Fig. 8c), which correlated with their revival frequencies in response to L-alanine (Fig. 8d).

To further corroborate the inference that the history of Ald expression in the progenitor cells controls spore outgrowth, we used the strain with the IPTG-inducible *ald* gene and induced Ald synthesis after the first late spores had formed (Fig. 9a). As expected, uninduced spores behaved like WT spores and a subset of early spores, but none of the uninduced late spores grew out. However, among spores derived from *ald*-induced progenitor cells, the capacity for spore outgrowth increased (Fig. 9b) and, despite their late onset of cell differentiation, these displayed the highest overall revival frequency (Fig. 9c).

**Memory generates a spore quantity vs. quality tradeoff**. To understand the implications of an intrinsically generated memory for the coupling of sporulation dynamics to revival success, we

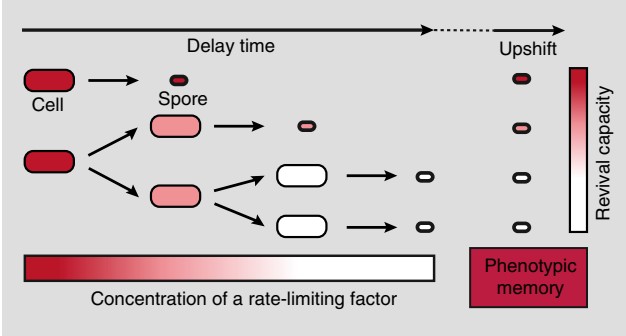

**Fig. 7** Intrinsic spore memory model. A factor X (red) limits spore revival and its production ceases during starvation. As cells delay sporulation, the concentration of X decreases due to dilution/degradation. Thus, less of X is available for carry-over into spores, resulting in spores with decreasing levels of X and differential revival capacities

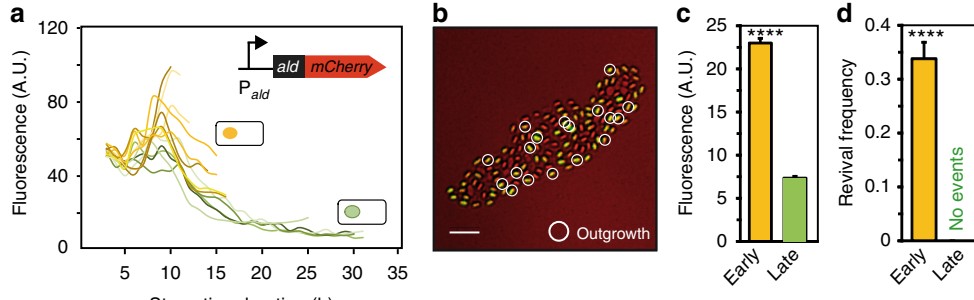

**Fig. 8** Spores obtain Ald-mCherry from their progenitors. **a** RFP fluorescence trajectories resulting from a Ald that was tagged with mCherry and that was expressed from the *ald* promoter (*amyE::P_ald*-ald-mcherry). The yellow (green) lines denote trajectories of cells that turn into early (late) spores. **b** Overlay of a bright-field (red) with a RFP fluorescence image (green) of the resulting spores. Scale bar: 5 μm. **c** Mean RFP fluorescence of early and late spores. **d** Corresponding revival frequencies in response to L-alanine. Strain: BIB1423. Data: mean ± SEM, $n_c = 8$ ($n_s ≥ 500$), unpaired *t* test: ****$P ≤ 0.0001$

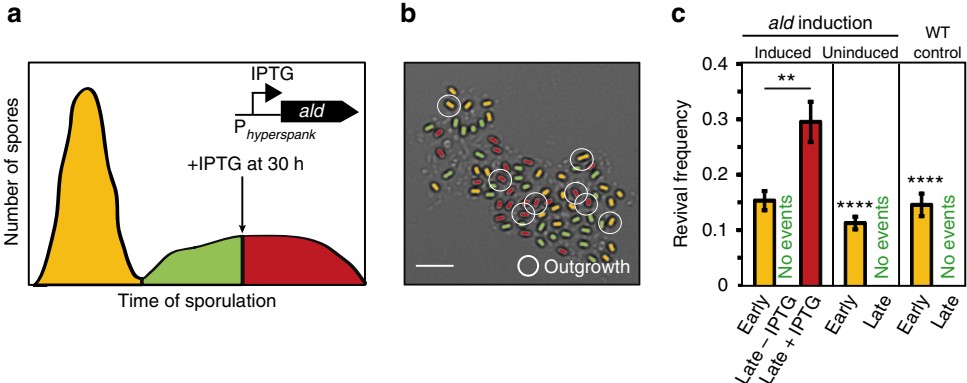

**Fig. 9** Alanine dehydrogenase contributes to a spore's intrinsic memory. **a** Schematics illustrating the formation of three different spore subpopulations by inducing Ald in the progenitor cells of spores formed after 30 h from an IPTG-inducible promoter. Scale bar: 5 μm. **b** Effect of Ald expression in progenitor cells on spore revival to L-alanine. Scale bar: 5 μm. **c** Corresponding revival frequencies. Strain: BIB1300, Data: mean ± SEM, $n_c \geq 13$ ($n_s \geq 1000$), unpaired $t$ test: **$P \leq 0.01$, ***$P \leq 0.001$, ****$P \leq 0.0001$. The data from the WT and the uninduced control was included as a reference

developed a mathematical model of sporulation and outgrowth. In the model, we assumed that the level of the controlling enzyme is set at the onset of starvation and thereafter declines owing to further cell divisions. Upon sporulation, the enzyme is stored in the spore (Fig. 7). With the help of a kinetic model for the sporulation dynamics, we analyzed the relationship between spore numbers and average enzyme level in the resulting spores as the sporulation rate was varied (Methods). We find that faster sporulation increases enzyme levels in the spores, thus raising spore "quality," whereas the spore yield ("quantity") will be low. Conversely, protracted dynamics of spore formation will yield a higher number of spores due to ongoing cell divisions, but these spores will be of lower quality due to their reduced enzyme levels. Thus, the model predicts a tradeoff between spore quantity and quality (Fig. 10a).

If this were indeed the case, a change in the signaling network that results in altered sporulation dynamics should not only affect spore yield but should concurrently alter the revival properties of the resulting spore population. To test this hypothesis, we interfered with signaling via the sporulation phosphorelay. Two central regulators of the delay-time distribution are the histidine kinase KinA and the RapA phosphatase, which activate and inhibit signaling, respectively. We used engineered strains in which the expression of either protein can be artificially controlled from an IPTG-inducible promoter (Fig. 10b). We then induced sporulation in the mutant strain in co-culture with the WT strain, and treated the resulting spores with L-alanine (Supplementary Movies 7 and 8). As expected, in the IPTG-induced KinA (RapA) strain, sporulation was accelerated (decelerated) relative to WT, which resulted in a decreased (increased) spore yield. Moreover, the mutants gave rise to higher (lower) fraction of outgrowing spores (Fig. 10c). We ruled out the possibility that induction of KinA per se was enhancing outgrowth (Supplementary Movie 9). To further corroborate our model, we repeated the experiments by varying the IPTG induction levels to fine-tune sporulation timing and also measured the Ald levels in the mutant spores using the fluorescence from the Ald-mCherry fusion protein as a proxy. As predicted, the spore yield and the average spore fluorescence are anti-correlated with each other (Fig. 10d), as are spore yield and the revival frequencies in response to L-alanine (Fig. 10e).

## Discussion

There is a growing body of evidence supporting the notion that cellular memory plays an important role in microbial stress responses. Effects resulting from carry-over of cellular components have been termed "phenotypic memory"[48]. In *Escherichia coli*, the *lac* operon shows a considerable capacity—based on phenotypic memory—to transmit metabolic information across several generations[49]. Furthermore, in *Bacilli*, not all proteins found in spores are synthesized during sporulation. Instead, spores obtain a substantial fraction of proteins from their progenitors, although little is known about their function in spore revival[23–25]. Our results show that alanine dehydrogenase is one such protein, and that it controls alanine-induced outgrowth. Presumably, as cells experience (alanine) starvation, Ald expression is turned down[47]. Protein levels are then progressively diluted as cells continue to divide, before being transferred into the spore. In this way, the capacity for spore revival is controlled by sporulation timing. Since many genes are turned off when bacteria experience a nutrient downshift, other genes probably contribute to overall spore memory in analogous ways. For the same reason, this principle should apply to the formation of an intrinsic memory in other spore formers[24]. On the other hand, not all revival traits that we found to be affected by sporulation timing must necessarily result from an intrinsic memory, as changes in environmental conditions during sporulation could also affect spore development[18]. Moreover, it will be interesting to investigate, whether other spore traits, such as spore resistance properties, are controlled by sporulation timing as well.

Theoretical models have predicted that survival strategies employing cellular memory could be beneficial[48–50]. In the *E. coli lac* operon, phenotypic memory results in a reduced lag time when a stimulus is repeated[49]. In *B. subtilis*, the transition from sessile to motile cells is controlled by memory, which might give cells sufficient time to explore the alternative lifestyle[51]. It is not yet known whether other types of memory, e.g., the history-dependent sporulation dynamics of *B. subtilis*[52] or the population-based "memory" in the *Caulobacter crescentus* salt-stress response[53], serve an adaptive purpose. Our data does not point to a direct benefit of spore memory. While the Ald-based memory resulted in differential spore outgrowth, it did not affect germination probability. Thus, late spores will irreversibly lose their protective properties upon germination, yet their impaired outgrowth prevents them from taking full advantage of the available nutrients. Moreover, we were able to overcome this outgrowth "defect" by inducing Ald prior to sporulation, which boosted the overall revival success of the population. We therefore favor the idea that phenotypic spore memory arises as a byproduct of evolution, and might be maintained by selective

pressures, e.g., costs of gene expression, not addressed by our experiments.

The fact that sporulation timing controls both the spore yield and the spore revival frequency argues strongly that the production of spores is subject to a quantity vs. quality tradeoff. Quantity vs. quality tradeoffs have been proposed as a

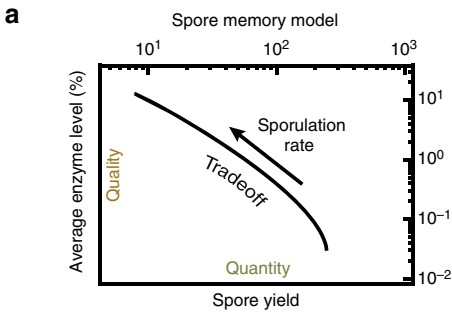

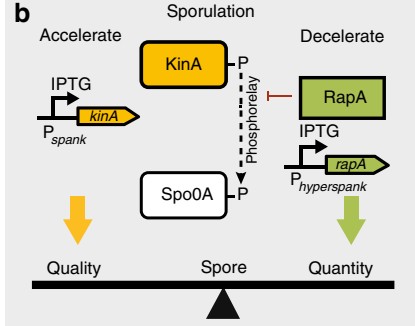

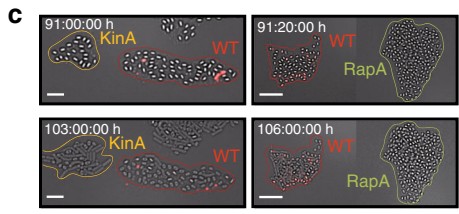

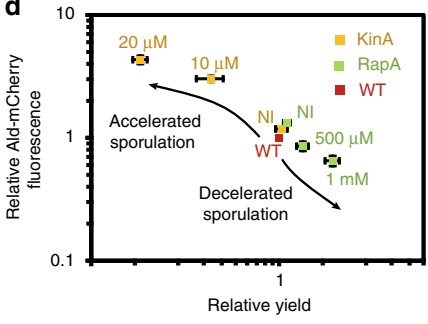

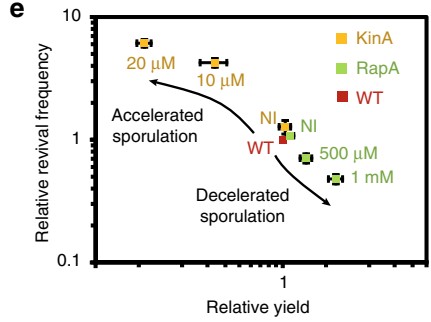

fundamental principle that shapes the production of reproductive units in response to resource limitations[54]. Examples include plant seeds and fish eggs[55,56]. Although, in bacteria, this tradeoff appears to be the result of phenotypic memory rather than constraints imposed by physicochemical factors during prolonged starvation, the consequences for evolution may be similar. In general, both the quantity and the quality of progeny determine the reproductive success of an organism. By analogy, for bacteria that survive fluctuating environments by cycling between a vegetative and a spore state, the success of spore revival is an important factor in determining their fitness. Under conditions where revival is limited by spore quality, rapid sporulation is favorable, whereas slower sporulation is beneficial when the yield determines revival success. Thus, irrespective of its evolutionary origin, an intrinsic spore memory could have an important evolutionary function by driving adaptations through quantity vs. quality tradeoffs.

For example, selective pressures on spore revival could favor the emergence of regulatory mutants of sporulation signaling. In our experiments, we mimicked essential features of the duplication of a gene for a sporulation kinase (or phosphatase). The increased (decreased) signaling activity in the sporulation phosphorelay accelerated (decelerated) sporulation in the mutants and thus enhanced spore quality (yield). Thus, duplications of genes for kinases or phosphatases or the acquisition of homologous genes by horizontal gene transfer could provide a selective advantage under particular upshift conditions (which may change over time). This observation could help to explain how diversification of the sporulation phosphorelay, which in *B. subtilis* is controlled by multiple kinases[57] and phosphatases[42], was favored during evolution[58]. We thus propose that quantity vs. quality tradeoffs contribute to the emergence of signaling complexity.

A similar line of thought suggests that, depending on the ecological niche, spore yield or spore quality may be more important for overall fitness. Interestingly, *B. subtilis* isolates from the gut of the broiler chicken exhibit accelerated sporulation compared to the soil-derived laboratory strain, because they lack several sporulation phosphatases[59]. We predict that this strain shows a decreased spore yield but generates spores that are capable of reviving more efficiently to weak nutrient stimuli, and we speculate that the gut environment may select for such spores. Hence, quantity vs. quality tradeoffs could contribute to niche adaptation of spore formers.

Complex environmental fluctuation patterns within a niche can lead to survival strategies that diversify the population to take advantage of bet-hedging. Our finding that early spores respond to weak nutrient stimuli more efficiently than do late spores suggests that environmental trajectories could exist, where accelerated sporulation provides a fitness benefit to *B. subtilis*. This might explain why early spores form under conditions in

**Fig. 10** Spore memory creates a quantity vs. quality tradeoff. **a** Results from a mathematical model showing a tradeoff between increasing spore yields (quantity) and decreasing average enzyme level per spore (quality) that emerges from memory. **b** To tip the balance between spore quantity and quality, KinA (RapA) was expressed from an IPTG-inducible promoter to accelerate (decelerate) sporulation. **c** Snapshots of *kinA* (BIB1334) and *rapA* (BIB1330) mutant colonies, that were co-cultured with a WT reporter strain (BIB1126). Images show spore colonies before (top) and after stimulation with L-alanine (bottom). Scale bar: 10 μm. **d** Fluorescence from Ald-mCherry vs. spore yield and **e** spore outgrowth vs. spore yield. Sporulation timing was modulated by the varying the IPTG levels in *kinA* (*rapA*) mutants. All values are normalized to the WT that was grown in co-culture in each case. Strains: WT (BIB1423), *kinA* (BIB1440), *rapA* (BIB1442), data: mean ± SEM, $n_c \geq 5$ ($n_s \geq 200$)

which vegetative cells can still replicate[14]. We thus propose that advancing sporulation in a subpopulation of cells in response to a nutrient downshift could be adaptive in light of uncertainty concerning the magnitude (or quality) of a future upshift by producing spores with higher quality. This could complement the bet-hedging with respect to uncertainty regarding the duration of the starvation period that is often invoked to explain why sporulation is delayed in another subpopulation of cells[13]. Variability in sporulation and spore revival is a trait found in many *Bacilli*[60]. Based on the above reasoning, a "first in–first out" relationship should be widespread among bet-hedging spore formers.

Finally, regardless of whether or not an observed complex behavior of a species is an evolved adaptive trait, such features often have tremendous practical implications for controlling microbial populations in industrial and medical settings. For example, variability in spore revival properties poses serious challenges for the sterilization of spore-contaminated food[18,60]. On the other hand, *B. subtilis* is a probiotic, and is administered to plants and animals in the form of spores. Thus, many protocols have been developed to increase the overall spore yield in the fermentation of industrially relevant strains[61,62]. We suggest that future applied research should take account of the possibility of an antagonistic relationship between spore yield and spore quality. Moreover, any given protocol might produce different sets of spores, which could show different propensities for revival in different niches. Hence, our findings might help to devise new protocols that enrich for specific spore types in the final products.

To summarize, our results provide a simple model for how phenotypic memory shapes bacterial dormancy. Cellular memory could thus link selection pressures acting at quite different times and life-cycle stages to produce unexpected connections between states. This could very well apply to other species and cellular states, and help us to understand and ultimately control complex microbial behaviors.

## Methods

**Vectors, plasmids, and strains**. All strains are derived from *B. subtilis* B168 (1A700). Genotype information on vectors (Supplementary Table 1), plasmids (Supplementary Table 2), strains (Supplementary Table 3), and primers (Supplementary Table 4) can be found in the Supplementary Information.

All vectors, plasmids and strains were constructed using standard molecular biological procedures. *E. coli* DH5α (Invitrogen, Carlsbad, CA, USA) was used for cloning. Luria broth (LB) agar plates were used to select transformants. When required, the appropriate antibiotics and amino acids were added as follows. For *E. coli*: ampicillin (100 µg ml⁻¹), for *B. subtilis*: chloramphenicol (5 µg ml⁻¹), kanamycin (10 µg ml⁻¹), spectinomycin (100 µg ml⁻¹), erythromycin (2 µg ml⁻¹), lincomycin (25 µg ml⁻¹), and tetracyclin (10 µg ml⁻¹).

**Vector construction**. To construct pRFP_Star, we amplified the backbone of pGFP_Star[63] (excluding GFP) with primers ST221 and ST162. *mCherry* was amplified from genomic DNA of *Bacillus* strain BIB182 (M. Elowitz, CalTech[9]) using primers ST219 and ST220. The two parts were fused by Gibson assembly[64]. Clones were checked by colony PCR with primers ST17 and ST243. Correct assembly was verified by analytical restriction digestion with SalI/SbfI and NdeI/SbfI. The vector region containing the T*gyrA* terminator, the site for ligation-independent cloning (LIC) and *mCherry* was sequenced with primers ST17 and ST243. The new vector has been made available to the Bacillus Genetic Stock Center (Columbus, OH, USA) under the accession no. ECE359.

**Plasmid construction**. *Fluorescent promoter fusions:* Fluorescent promoter fusions were constructed using the pXFP_Star vectors[63] by ligation-independent cloning (LIC)[65]. Promoters P*spoIIE* (EIB404) and P*ald* (EIB450) were amplified from *B. subtilis* 168 genomic DNA using the indicated primers in Table 4 and the fragments were cloned into pGFPamy_Star by LIC. P*rapA*-*mCherry* (EIB432): the regulation of *rapA* transcription is very complex and includes a CodY repressor site that is located in the *rapA* gene[66]. To construct a transcriptional reporter that contains all known regulatory elements, upstream and downstream parts of the *rapA* promoter were amplified from *B. subtilis* 168 genomic DNA using primers ST1 and ST222 (up-fragment) as well as ST223 and ST224 (down-fragment). The down-fragment was designed to exclude the native ribosome binding site of the *rapA* gene to prevent translation of undesired fusion proteins. The two parts were

fused by overlap extension PCR and cloned into pRFP_Star by LIC. All inserts were verified by sequencing with primers ST17 and ST243.

*Ald-mCherry reporters:* EIB503: P*ald*-*ald* was amplified from *B. subtilis* 168 genomic DNA using primers MA55, MA56, and *mCherry* from plasmid EIB422 using primers MA57, MA58, respectively. The two fragments were fused by overlap extension PCR using end primers MA55 (KpnI) and MA58 (BamHI). The resulting fusion fragment P*ald*-*ald*-*mCherry* was ligated to the multiple cloning site of pSac-Kan[67] by restriction enzyme ligation cloning (RELC). The insert was sequenced with primers ST283 and ST284.

EIB499: P*ald*-*ald*-*mCherry* was ligated into the small cloning site of EIB450 by RELC. This resulted in a double reporter (P*ald*-*gfp* P*ald*-*ald*-*mCherry*) that can be used for correlating *ald* promoter activity to Ald-mCherry levels. The inserts were sequenced using primers ST16, ST17, and MA58.

*Induction of protein expression:* pDR111-*rapA* (EIB297): The *rapA* coding region was amplified from *B. subtilis* 168 1A700 genomic DNA using SS34 (NheI) and SS37 (SphI) primers and ligated to the multiple cloning site of the pDR111 vector (D. Rudner, Harvard University, Boston) by RELC. The insert was sequenced using primers SONSEQ18 and SONSEQ19.

pDR111-*ald* (EIB452): The *ald* coding region was amplified from *B. subtilis* 168 1A700 genomic DNA using MA37 (NheI) and MA38 (SphI) primers and ligated to the multiple cloning site of pDR111 vector by RELC. The insert was sequenced using primers SONSEQ18 and SONSEQ19.

pSac-KAN-P*hyperspank*-*ald* (EIB480): The region containing "P*hyperspank*-*ald* and *lacI*" was amplified from EIB452 DNA using MA47 (SpeI) and MA48 (SacI) primers and inserted in to the multiple cloning site of pSac-KAN vector by RELC. The plasmid was sequenced with primers ST283, ST284, MA49, and MA50.

**Strain construction**. Strains were obtained by transforming *B. subtilis* with the indicated ectopic integration vectors as summarized in Supplementary Table 3 using standard protocols[44]. All fluorescent reporters are present in single copy in the chromosome and are expressed from an ectopic locus (*amyE*, *sacA*, *pssB*). If not indicated otherwise, correct locus integration was verified by a locus PCR and the absence of a single cross-over excluded by PCR using appropriate primers given in Supplementary Table 4.

P*trpE\**-*mCherry* P*spoIIE*-*gfp* (BIB1019): The P*trpE\**-*mCherry* cassette, including the erythromycin resistance cassette and flanking regions for homologous recombination with the *ppsB* locus of *B. subtilis*, was amplified with primers ST133 and ST134 from genomic DNA of BIB182 (Michael Elowitz, CalTech), gel purified, and transformed into BIB224 resulting in strain BIB444. Correct locus integration was verified by PCR. BIB444 was subsequently transformed with EIB404 (P*spoIIE*-*gfp*). Transformants were screened for loss of amylase activity on LB starch plates (1% w/v final). Clones were verified for *amyE* locus integration by PCR with primers ST39 and ST40 and absence of single cross-over events by PCR with primers ST129 and ST130. The promoter sequence was also checked by sequencing with ST16 and ST17.

The Δ*ald* knockout strain (BIB1416) was constructed via transformation of BIB444 with a *ald::tetR* fusion product. *ald*-up, *ald*-down fragments amplified from *B. subtilis* 168 1A700 genome with MA51, MA52 and MA53, MA54 primer pairs, respectively. *tetR* cassette was amplified from the pDG1514 vector[68] (Supplementary Table 1) with LA54 and LA55 primers. The resulting fragments were fused with overlap extension PCR using the end primers MA51 and MA54. Transformants were checked via colony PCR with LA58, KN1, KN3, and KN4 primers.

**Media and solutions**. Strains were grown in Luria–Bertani broth (LB), CH medium, SM[69] or SM* (i.e., SM with a reduced level of glutamate (10%) and 1 mM L-alanine) at 37 °C with aeration, unless otherwise noted. Tryptophan was added at 20 µg ml⁻¹ and 22 µg ml⁻¹ for CH and SM/SM* media, respectively. Nutrient solutions for spore revival experiments were performed with L-alanine (100 mM) and AGFK (19.8 mM L-asparagine, 33.6 mM D-glucose, 33.6 mM D-fructose, and 60 mM KCl). Inducer solutions for protein expression: 10 mM isopropyl-β-D-thiogalactopyranosid (IPTG).

**Life history assay**. *Induction of sporulation:* Overnight cultures were inoculated from single colonies grown on selective LB plates and incubated at 37 °C (with selection). Cells were diluted to a starting OD₆₀₀ nm = 0.015 in fresh CH media and grown to an OD₆₀₀ nm = 0.8–1.0. Cells were centrifuged and re-suspended in SM* media at an OD₆₀₀ nm = 0.1. Aliquots (4 µl) of the cells were spotted onto 1.5% ultrapure agarose (Invitrogen) agarose pads containing SM* media (without selection). For experiments involving the co-culture of two strains, strains were grown separately and then mixed in a ratio of 1:1 before loading the pad. Defined agarose pads were cast using eight-well (9 mm diameter, 1 mm deep) press-to-seal silicone insulators (Molecular Probes) as a mold, which was sandwiched between two coverslips. Once the pads had dried, the pads were inverted and stamped into a 24-well glass-bottom SensoPlate (Greiner Bio-One). To prevent drying of pads during extended imaging, the space between the wells was filled with dH₂O to create a humid environment. The edges of the plate were covered with silicon paste (Baysilone, Bayer) and the plate was sealed with parafilm (two to three-fold). The starvation response was then monitored by time-lapse microscopy as described below.

*Induction of spore revival:* After 90 h of starvation (measured from the time of resuspension in sporulation media), the SensoPlate was removed from the microscope. To induce spore revival, 10 µl of the required solution was applied on top of the agarose pad in each case. Nutrients then diffuse into the pad to reach the spores at the bottom. Thus, spores are estimated to experience an effective concentration of about 1/6th-1/7th that of the stock solution, assuming that the nutrients equilibrate sufficiently fast in the pad ($V_{pad}$ ~ 60 µl) before being consumed by the reviving spores. After addition of the nutrient solutions, the plate was re-sealed and placed back into the microscope. The position of the plate/stage was adjusted to identify the colonies that had been imaged during the downshift period, and imaging was resumed and continued for 12 h to monitor spore revival.

*Induction of protein expression:* To temporally control protein expression during time-lapse experiments, we applied the same procedure as for nutrient induction of spores. At the indicated time, 6 µl of IPTG was pipetted onto the back of the agarose pad. For induction of $P_{spank}$-*kinA* and $P_{hyperspank}$-*rapA* strains, SM* pads were supplemented with 10, 20, 500, and 1000 µM of IPTG (final concentration in the pad), respectively.

**Microscopy.** Cells and spores were imaged using an automated DeltaVision Elite Imaging System (Applied Precision, Issaquah, WA, USA) equipped with an Olympus IX71 inverted microscope, the Ultimate Focus system and an environmental control chamber kept at 36.5 °C. Samples were imaged with a 40x/NA = 0.95 air objective (Olympus 1-U2B828) and a 1.6-fold auxiliary magnification lens. Bright-field imaging was performed with a condenser with a long working distance (Olympus IX2-LWUCD). Fluorescent reporters were excited using the fluorescent protein Insight solid-state illumination unit with the following settings and filter sets (excitation, emission): GFP: 0.1 s with 100% excitation (475 nm/28, 523 nm/36), mCherry: 0.1 s with 100% excitation (575 nm/25, 632 nm/60), and a Quad dichroics (reflection bands: 381–401 nm, 464–492 nm, 561–590 nm, 625–644 nm; transmission bands: 409–456 nm, 500–553 nm, 598–617 nm, 652–700 nm). The ultraviolet filter was engaged to minimize phototoxicity. Multiple stage positions were monitored with a motorized microtiter stage at a frame rate of once in every 20 min unless noted otherwise. Images were recorded with a PCO Edge sCMOS camera using 1024 × 1024 pixels. Automated imaging was performed by an Instrument Controller with Resolve3D SoftWorx-Acquire Version 6.1.1 Release 5.

**Experiments using spores collected from well-mixed cultures.** Cells were first grown in CH medium as described above, then re-suspended in 5 mL SM medium at an $OD_{600nm} = 0.5$ and shaken for 4 days in a culture tube at 37 °C. Next, the spores were washed and re-suspended in phosphate-buffered saline (PBS). For spore-aging experiments, the collected spores were stored at room temperature in PBS. Spore revival assays were performed at the indicated time with respect to the day of spore harvest. For the revival assays, 4 µl of the spore suspension were placed on an agarose pad containing SM medium with a reduced level of glutamate (10%). SM pads support neither spore outgrowth nor germination. Spore revival was then induced by the addition of the indicated nutrient solutions to the agarose pads, and the response was monitored by time-lapse microscopy as described above. Spores generated in liquid SM and placed on fresh SM pads were in general much more responsive to nutrient stimuli than spores that had been generated on agarose pads and were induced on used SM* pads. In order to reveal differences between the spore subpopulations, nutrients were applied at 100-fold dilutions. To test the effect of heat activation on revival capacity, spores were exposed to a temperature of 80 °C for 20 min, prior to L-alanine stimulation.

**Mathematical model.** We consider cells that contain an enzyme *e* which is required for spore revival. Upon nutrient downshift at time $t = 0$, cells stop enzyme production but continue to grow at a rate $\lambda$, so that the normalized enzyme concentration in every cell decreases as $e(t) = \exp(-\lambda t)$. Cells also progress at rate $\eta$ from their naive state $c_0$, through intermediate states $c_1,\ldots,c_k$, toward the starved state $c_s$, where they may die at rate $\mu$. Cells may sporulate from all states $c_1,\ldots,c_k, c_s$ with rate $\delta$. Sporulation at time *t* generates a spore denoted as $s^{e(t)}$, which conserves the current enzyme level $e(t)$. The dynamical equations for the state populations are:

$$\partial_t c_0(t) = (\lambda - \eta)c_0(t); \partial_t c_i(t) = \eta c_{i-1}(t) + (\lambda - \eta)c_i(t) \text{ for } i>0;$$
$$\partial_t c_s(t) = \eta c_k(t) + (\lambda - \mu)c_s(t);$$
$$\partial_t s^e(t) = \delta[e - e(t)] \times \delta \times \left\{ \sum_{i=1}^{k} c_i(t) + c_s(t) \right\}.$$

$$(1)$$

We solve these equations for $k = 2$ with $c_0(0) = 1$ and all other populations 0 as initial condition. The spore quantity or spore yield is the number of spores generated per initial cell, $s = \lim_{t\to\infty} \int_{e=0}^{1} s^e(t)de$. The spore quality is defined by the average enzyme level per spore, $\bar{e} = \int_{e=0}^{1} e\rho(e)de$, where $\rho(e) = \lim_{t\to\infty} s^e(t)/s$ is the distribution of enzyme levels in the generated spore population.

**Image analysis.** A combination of ImageJ plugins (ImageJ; http://imagej.nih.gov/ij/) and custom-written MATLAB programs were used to analyze microscopy data

as described below. Fluorescence trajectories were obtained by tracking individual cells and subtracting background fluorescence[70].

**Quantification of spore revival traits.** We counted the number of dormant spores $N_s$, the number of outgrowing spores $N_o$, and the number of germinated spores $N_g$ in each microcolony after a specified time. We adjusted the time interval to the maximum possible value at which we could faithfully track the spores for each media condition. A spore was classified as "germinated" when its refractivity had dropped substantially (>30%) and as "grown out" when the spore was visibly cracked open. We used the following definitions to determine the revival frequency $f_r = N_o/N_s$, the germination frequency $f_g = N_g/N_s$, and the outgrowth frequency $f_o = N_o/N_g$. The spore yield *Y* was defined as $Y = N_s/A_c$, where $A_c$ is the initial segmented area of the starving microcolony after the shift-down. Spore quality *Q* was determined by measuring the fluorescence intensity of spores from Ald-mCherry. For each condition, a WT control was included and used to determine relative values for *Y* and *Q* by normalizing to the WT in each case.

**Statistical analysis.** Results report the averages from $n_c \geq 4$ microcolonies (movies) from $n_e \geq 2$ independent experiments, which were typically run using two independent clones, resulting in at least $n_s \geq 200$ tracked spores. Error bars denote the SEM. Unpaired *t* test was used to determine the statistical significance of the observed differences (GraphPad; http://www.graphpad.com). The number of stars indicates the *P* value with *$P \leq 0.05$, **$P \leq 0.01$, ***$P \leq 0.001$, ****$P \leq 0.0001$.

**Code availability.** Custom-made code for image processing can be obtained from the corresponding author upon request.

**Data availability.** All relevant data supporting the findings of the study are available in this article and its Supplementary Information files, or from the corresponding author upon request. Supplementary Movies are also available from the AV-portal of the Technische Informationsbibliothek (https://av.tib.eu/upload) under http://doi.org/10.5446/33993, http://doi.org/10.5446/33994, http://doi.org/10.5446/33995, https://doi.org/10.5446/34001, http://doi.org/10.5446/33996, http://doi.org/10.5446/33997, http://doi.org/10.5446/33998, http://doi.org/10.5446/33999, http://doi.org/10.5446/34000.

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

## Acknowledgements

We thank Isabel Martin, Lilija Aprupe, and Charlotte Kaspar for help with image analysis, Heiko Babel for help with cloning, Katja Bettenbrock for discussions, and Paul Hardy for critical reading of the manuscript. We thank O. Kuipers, M. Elowitz, and D. Rudner for sending strains and plasmids. This work was funded by an ERC Starting Grant (GA 260860) from the European Research Council and an Emmy Noether Grant (BI1213/3–1) from the Deutsche Forschungsgemeinschaft (DFG) awarded to I.B.B. K.R. acknowledges support by the Federal Ministry of Education and Research (BMBF) via ImmunoQuant (e:Bio). T.H. and N.B. were supported by EraCoSysmed (OPTIMIZE-NB). I.B.B., K.R., and T.H. are CellNetworks members. K.N. was supported by a Cell-Networks postdoc fellowship. I.B.B. also acknowledges support from the Priority

Program SPP1617 funded by the DFG, the Large Scale Data Facility (INST35/931−1 FUGB) supported by the DFG, and the Ministry of Science, Research and Arts Baden-Württemberg (MWK) and support by the Max Planck Society.

## Author contributions

I.B.B. designed the research, A.M., S.T., and K.N. constructed the strains, A.M. and M.Z. performed the experiments, J.-P.B., I.B.B., and K.R. developed methods and software for image analysis, A.M. and I.B.B. analyzed the data, N.B., T.H., and I.B.B. developed the mathematical model, and I.B.B. wrote the manuscript with help from all the authors.

## Additional information

**Competing interests:** The authors declare no competing financial interests.

