## [Peer Review File · Nature Communications]

Reviewer #1 (Remarks to the Author):

In this interesting, innovative and well-written paper, the authors show that a population of spore-forming *B. subtilis* cells has a diverse response to germination signaling. The timing of germination seems to be related to events that transpired during sporulation. The movies showing the behavior of individual cells in small clusters are particularly impressive. However, the authors' terminology and its implications are not conventional in the microbiology field and, in this reviewer's opinion, misrepresent the actual events.

Specific comments:

- 1) Title and Abstract: "Memory" seems like an attractive term, but it isn't evident that it has any substantial meaning in this case. If the way in which cells and spores function is a reflection of the intracellular concentrations of specific compounds, why not just say that, instead of implying that the bacteria have human-type behavior? (Of course, human memory isn't fully understood either.) Similarly, referring to the intracellular concentration of an enzyme or signal as "phenotypic memory" is enigmatic rather than explanatory.
- 2) Fig. 1: How long do GFP and m-Cherry persist? Unless they are very labile, the sporulating cells should contain both. In cells that express only *trpEp-mCherry*, when does the mCherry fluorescence disappear?
- 3) Line 112: The sentence is misleading in several ways. The pre-spore is neither a dormant spore nor an early sporulating cell.
- 4) Line 115: Does "replicate" mean "divide" or "double in mass"?
- 5) Line 117: The timing of spore formation in the method used is relatively slow. In liquid medium, pre-spores would appear about 4 hrs after shifting to the Sterlini-Mandelstam medium.
- 6) Line 119: How was "grow at a slow rate" measured?
- 7) Line 135: Germination is also rather slow under the conditions used.
- 8) Fig. 2: Why do so few spores germinate under any of the conditions tested? Normally, germination is >90%. The spores are not really in CH or LB media, but rather exposed to a 6-fold dilution of these media. The legend and figure disagree as to whether one of the media is GFAK or AGFK.
- 9) References 19 and 31: The claims of these papers may not be entirely correct. A recent paper (Korza et al. 2016. *J Bacteriol.* 198:3254-3264) challenged some of their most important conclusions.
- 10) Fig. 4C and line 250: Was the last sample taken at 120 days (4 months) or at 90 days (3 months)?
- 11) Fig. 5: How much time was allowed for maturation of the "late" spores (completion of dormancy)? If one waited longer, would the "late" spores have germinated faster?
- 12) Line 409: RapA is not a repressor of *kinA*. RapA counteracts the effect of KinA.
- 13) Lines 466-474: Use of the word "memory" here isn't explanatory. Ald is present at variable concentrations; it isn't a question of whether the cell remembers that it has a certain concentration of Ald.

- 14) Line 520: The presence and activities of the different kinases are a response to different environmental conditions.
- 15) Line 528: It isn't clear what "more efficient spores" means. Perhaps the intent is to refer to cells that sporulate more efficiently.
- 16) Line 552: It isn't clear what "quality" means. From the bacterium's point of view, quality would refer to the level of resistance to heat, chemicals, etc.
- 17) Line 580: Would addition of a 1/6 volume of CH or LB induce germination in a test tube?
- 18) Line 591: Change "stop" to "slow down".
- 19) Line 622: How much swelling was taken as an indication of the onset of germination?
- 20) Supplementary material: The movies are exceptional. But it is unclear what the actual conditions are. Is it correct that the medium for the agarose on which the cells were placed is SM medium with reduced glutamate? Why was L-alanine added to the medium? If the same cells were used to inoculate a liquid culture at the equivalent concentration in the same medium, how quickly would sporulation occur and what fraction the cells would sporulate?

Reviewer #2 (Remarks to the Author):

This ms. is directed at a central question of microbiology: how does the phenotypic diversity of single cells affect their future behavior. In specific, the ms. reports observations regarding the effect of sporulation timing on the ability of dormant spores to germinate. The study uses novel methodological innovations that allow for the first time an analysis of the germination behavior of single spores and their developmental history that provide new insights into the system. In addition, their use of modeling to make explicit predictions is impressive and will be a great example for how modeling is useful for understanding biological systems. The ms. is well-written and data convincing. I do however, have several issues that the authors should consider.

1. Why are the times for the various germinants different in Fig, 2D? Also, why do the authors think that there is only ~2-fold effect with LB, much different than the absolute effect of CH or L-ala? That is, why should different germinants act qualitatively differently?
2. I find Fig 4A confusing – this distribution is not really "bi-modal" (line 228) – and what is the cutoff? How did they choose it?
3. Why do the effects they see with the RapA reporter (low rapA vs. high rapA) on germination (Fig. 4B) not as strong as (Fig. 2B; early vs. late spores)
4. While the author's talk quite a bit about a gfp fusion to ald promoter as a reporter for Ald levels (e.g., line 332), they do not present any data that there is a correlation between Ald protein levels and reporter activity. In addition, they state that the decay in the reporter is reflective of a decrease in expression from the ald promoter (line 335) – but in the absence of any data regarding the stability of the reporter under these conditions, a direct correlation is only an assumption
5. The posit that Ald levels in "early" spores are higher than those of "late" spores (line 336-8). They could test this assumption by using FACS to isolate the separate populations and determining the relative Ald levels.
6. They should test that the ald induction (Fig. 6D) affects alanine-dependent germination specifically (e.g., try other germinants).
7. Although their modeling in Fig. 7 is very intriguing, it would be strengthened if they could provide more than a single point (Fig. 7D) for KinA and RapA induction; a full dose-response curve

is not necessary but a second point for each would greatly increase confidence that they are on the right track. They could also try the Bs natural isolates (line 527) that apparently exhibit accelerated sporulation to see how they fit on the curve.

Minor points

Line 149 "late" should be italicized

Line 512 "driving" should not be italicized.

-Jonathan Dworkin

Reviewer #3 (Remarks to the Author):

In this manuscript the authors demonstrate that after an asynchronous sporulation, the spores that germinate relatively early are derived from cells that completed sporulation relatively early. This is an interesting observation. However, there are several significant difficulties. First, as detailed below, several experiments are compromised by a misunderstanding of the role of L-alanine, which does not support outgrowth. This has important implications for the authors' interpretation of some experiments, which are not convincing. There are other difficulties with experimental interpretation. Importantly, the manuscript is quite difficult to read in some places. This includes the use of informal terminology. Overall, some important interpretations and claims of the paper are not convincing.

There is no acknowledgement of extensive past work on variation in sporulation and germination (particularly from the Setlow lab) is made. This is important contextual information and it is surprising that this work is not discussed.

Specific comments:

The abstract is very difficult to understand. This is because of the use of obscure terminology and phrases. Examples of obscure terminology include "Phenotypic memory", "imaging life histories" and "history-labeling" (26-27), which are only defined later on. Obscure phrases include "sporulation timing", "kinetically favored", "early spores" and "late spores" (27-28). More importantly, the abstract does not provide any experimental data or specifics and, as a result, does not do an adequate job of conveying the real nature of the results obtained, or the empirical basis of any of the claims.

50: "phenotype" is the wrong word.

59: what does "state transition" refer to?

61-62: it is essential here to avoid making an unsupported claim; bet-hedging per se has never been demonstrated. Rather, the idea that these behaviors are a type of bet-hedging (and, therefore, adaptive) remains a speculation (although an intriguing one).

73: What does heterochronous mean? Asynchronous? If so, use that more conventional word.

87-88: The text should not give the implication that this sort of microscopy is novel; numerous

studies have visualized sporulation and germination dynamics in single cells.

129: This is an important point; it is well established that L-alanine is not at all a nutrient. This matters to experimental interpretation. By "revival response", do the authors mean "outgrowth"? Here and elsewhere, switching between terms in this way and, especially, the use of informal terminology, leaves reader confused not only as to the specific assertions being made, but also regarding the overall logic.

At an intuitive level, I understand the reason for the choice of the time point used to distinguish between early and late spores. However, this choice is not explicitly justified at this point, and it's not really clear that the conclusions of the paper are insensitive to the choice. What are the results of a sensitivity analysis to address this question? Specifically, how would this analysis affect figure 2D?

153: How is revival frequency defined?

176-177: I don't see how heat activation is relevant to the point being made.

180: This does appear to provide good support for the earlier time choice. However, the data in figure 3C showing the fluorescence changes at about 20' are from a limited data set, and it's not entirely clear what we are looking at. Is this the combined data from two experiments, or two separate experiments that are, somehow, averaged? I think this matters, as the apparent break point within the data could be a fluctuation. If it is real, this would be interesting, but needs a better statistical validation.

261 and following: I don't see how the experiments immediately prior to this conclusion strongly support the conclusion. I think the conclusion is a reasonable working model, and a better likely explanation for the phenomenon, but this is largely because it is already clear from earlier literature that sporulation results in variation in the spore population. But I don't see how alternative explanations are specifically excluded.

277: How was swelling measured? It appears that refractility was measured, not swelling.

279: It is imprecise to say that germination was faster. It appears that the completion of the conversion to the phase bright state was faster.

301 and following: I am unclear on the precise experimental question being asked. But, if the experiment hinges on L-alanine stimulating outgrowth, then the experiment is flawed, since L-alanine does not have this effect.

324 and following: I don't think this experiment demonstrates what is claimed, for the reason already cited and also because the presumption on line 335 is not compelling. Certainly, metabolism may depend to some degree on alanine. However, alanine alone does not support outgrowth, and the involvement of the spent medium in the effects seen here prevent a clear interpretation.

342: Given the potentially pleiotropic effects of Ald, it is hard to see how one would uniquely interpret these data.

348-350: This is very confusing. By differentiation, do the authors mean sporulation? If so, is this a post-hoc analysis, looking retrospectively at cells that ultimately became what were shown to be early spores? I am unable to follow the description of the experiment.

395 and following: I am not sure how useful the model really is. It is a dynamical description of the authors' interpretation of the data, but I don't think that this representation of the dynamics

actually provides any support for any conclusion.

404 and following: It is difficult to be convinced by the interpretation put forward by the authors. The effects of the mutations documented here could have many interpretations, and don't seem to provide support for a unique interpretation.

Response to the Referees

We would like to thank the reviewers very much for dealing with our manuscript and for the constructive comments provided. The revised manuscript was modified in accordance with the referees' comments and the editorial guidelines. In particular, we have conducted experiments to answer the questions raised by referee 1. We performed a comprehensive set of additional experiments to strengthen our model by following up the suggestions of referee 2 and addressed the concerns of referee 3 with the help of additional control experiments. Furthermore, we followed the suggestions of all referees to improve the clarity of the manuscript and streamlined the presentation to comply with the format requirements of the journal.

The most significant changes to the manuscript are summarized below:

- The abstract, introduction and discussion were revised to introduce and define the term spore memory.
- The relationship between spore revival, germination and outgrowth is clearly described in the Results section, and data on germination and outgrowth in response to L-alanine is introduced earlier in the manuscript (the original Figure 5 is now Figure 3).
- The classification of spores based on fluorescence from the P_{rapA} reporter was validated using larger sample sizes (Figure 4C). In Figure 5 the median fluorescence was used to partition spores into early and late classes.
- The mechanistic analysis of the action of Ald is supported by additional data from an *ald* knockout strain. In co-culture with a wild-type strain, the mutant sporulates and germinates normally (Supplementary Video 5). However, the mutant spores show impaired outgrowth in response to L-alanine (Extended Data Figure 5). This nicely complements our Ald induction experiment (Extended Data Figure 6). Notably, perturbation of Ald levels has no effect on the revival response to AGFK (Extended Data Figure 7). Together, these data suggest that, under our experimental conditions, Ald specifically controls alanine-induced outgrowth. All these data are now presented in a separate section "*Alanine dehydrogenase controls alanine-induced outgrowth*" before considering the data on an Ald-based spore memory.
- The spore memory model is supported by additional data that address the levels of the relevant protein, Ald. To this end, we studied the behavior of a reporter strain expressing an Ald-mCherry fusion protein under the control of the *ald* promoter. We find that fluorescence is carried over into the spore, and that early spores have higher fluorescence levels than late spores (Supplementary Video 6, Figure 6B).

- The predicted emergence of a quality-quantity trade-off is supported by additional data points. Moreover, we demonstrate that mutant spores contain higher (lower) levels of Ald (by using the mCherry fusion protein as a proxy) when sporulation is accelerated (decelerated). These data are summarized in two panels in Figure 7d. We also conducted a control experiment to exclude the possibility that induction of KinA *per se* might alter the spore revival response to L-alanine (Supplementary Video 9).

These changes have been highlighted in the revised manuscript.

Please find our point-by-point responses to the Referees' comments below. We hope that the revised manuscript is now acceptable for publication in *Nature Communications*.

Reply to the comments of referee 1

C1: In this interesting, innovative and well-written paper, the authors show that a population of spore-forming *B. subtilis* cells has a diverse response to germination signaling. The timing of germination seems to be related to events that transpired during sporulation. The movies showing the behavior of individual cells in small clusters are particularly impressive. However, the authors' terminology and its implications are not conventional in the microbiology field and, in this reviewer's opinion, misrepresent the actual events.

R1: We thank the referee for these favorable comments on our work and for pointing out where our presentation requires improvement. The specific comments have been very helpful and are addressed below.

C2: Title and Abstract: "Memory" seems like an attractive term, but it isn't evident that it has any substantial meaning in this case. If the way in which cells and spores function is a reflection of the intracellular concentrations of specific compounds, why not just say that, instead of implying that the bacteria have human-type behavior? (Of course, human memory isn't fully understood either.) Similarly, referring to the intracellular concentration of an enzyme or signal as "phenotypic memory" is enigmatic rather than explanatory.

R2: We agree that here the memory is "encoded" simply in protein concentrations and we understand the concern of the referee that the term "memory" could be wrongly associated with human-type behavior. Our wording was motivated by the term "phenotypic memory", introduced by Jablonka et al. in a 1995 study of the advantages of "carry-over" of cellular components from one generation of cells to the next. Since then this term has been used in numerous publications, including work on bacteria. Also in the physics literature, the term "memory" is conventionally used to describe any kind of dependence of behavior on prior history. In the revised manuscript we define "memory" in the Abstract. We have also revised the Introduction and the Discussion to provide context for this terminology and clearly differentiate it from human-type memory.

C3: Fig. 1: How long do GFP and m-Cherry persist? Unless they are very labile, the sporulating cells should contain both. In cells that express only trpEp-mCherry, when does the mCherry fluorescence disappear?

R3: For both GFP and mCherry we observe a drop in fluorescence when the pre-spore is released from the mother cell. The reason for this is not known. Fluorescence in the spores remains constant thereafter. mCherry is affected to a lesser extent than GFP. Notably the auto-fluorescence from the medium is stronger in the green than in the red

channel. Using an appropriate intensity scaling, a red fluorescence signal could be clearly detected in spores from a P_{trpE} -mCherry strain, while a GFP signal from spores is more difficult to detect above the background in our experiments.

C4: Line 112: The sentence is misleading in several ways. The pre-spore is neither a dormant spore nor an early sporulating cell.

R3: We agree that the sentence was misleading. It was erased and the reference to Fig. 2A is made later in the text.

C5: Line 115: Does “replicate” mean “divide” or “double in mass”?

R5: We understand “replicate” to mean “make two copies from one”, thus it implies both division and biomass growth. The sentence now reads: *Following the nutrient downshift, cells at first continued to divide and grew into small microcolonies.*

C6: Line 117: The timing of spore formation in the method used is relatively slow. In liquid medium, pre-spores would appear about 4 hrs after shifting to the Sterlini-Mandelstam medium.

R6: We agree that spores would appear earlier in liquid medium. However, in similar time-lapse experiments on agarose pads performed by other labs the onset of sporulation occurs around the same time as in our experiments. See for example Levine et al., PloS Biology, Figure 1D.

C7: Line 119: How was “grow at a slow rate” measured?

R7: There is both growth and cell division. We typically measure division rates. We replaced “grow” by “divide”.

C8: Line 135: Germination is also rather slow under the conditions used.

R8: The impression that germination was slow may be an issue of misleading wording. The statement that “*germination was complete within 2 hours*” was intended to say that we did not observe any additional germination events after this time. Indeed many spores have germinated already in the first frame after the upshift (see Supplementary Video S1). As show in Figure 3 (previous Figure 5) changes in refractive properties are visible already after 5 minutes, which is comparable to previous observations, see e.g. Zhang et al., 2010. To improve clarity we re-phrased “*Under our conditions, following stimulation with L-alanine, most spores germinated (>90%). However, only a subset of spores in each microcolony grew out to complete spore revival by the end of our 12-h observation period.*”

C9: Fig. 2: Why do so few spores germinate under any of the conditions tested? Normally, germination is >90%. The spores are not really in CH or LB media, but rather exposed to a 6-fold dilution of these media. The legend and figure disagree as to whether one of the media is GFAK or AGFK.

R9: We agree that germination levels at high levels of nutrient germinants typically exceed 90%. Indeed, in response to high concentrations of L-alanine more than 90% of the spores germinated under our conditions (see Figure 3 and Extended Data Figure 1). Figure 2 scores the *revival frequency*, defined as the fraction of all spores that grow out successfully. We also agree that spores are exposed to a dilution of the indicated media. We deliberately chose challenging nutrient conditions to ensure that only a small subset of spores will be able to grow out. To improve clarity we now define the terms *revival frequency*, *germination frequency* and *outgrowth frequency* at appropriate positions in the main text and include more details in the Methods section. Moreover, we introduce the data on germination and outgrowth in response to L-alanine earlier in the manuscript (Figure 5  Figure 3). In the legend of Figure 2, we refer to the Methods section when describing the ‘types of nutrients’ that were added to the agarose pad and have corrected GFAK to read AGFK.

C10: References 19 and 31: The claims of these papers may not be entirely correct. A recent paper (Korza et al. 2016. J Bacteriol. 198:3254-3264) challenged some of their most important conclusions.

R10: The fact that the properties of a spore can change after it has been released from the mother cell provides important contextual information for our study. The work by Korza et al. agrees with the previous studies on this point. The controversy as to whether translation is part of germination - which was nicely discussed by Boone and Driks - is not relevant for the present work. We now include the reference to the work by Korza et al. as suggested by the referee.

C11: Fig. 4C and line 250: Was the last sample taken at 120 days (4 months) or at 90 days (3 months)?

R11: Thanks! It was taken at 120 days. The typo was corrected.

C12: Fig. 5 (now Fig.3): How much time was allowed for maturation of the “late” spores (completion of dormancy)? If one waited longer, would the “late” spores have germinated faster?

R12: Most late spores have been released from the mother cell after 2 days. We waited for 4 days before inducing germination. Moreover, experiments performed on spores kept in

buffer for up to 120 days did not provide any evidence that extending the interval prior to the upshift changes the behavior of the late spores (see Fig. 5c).

C13: Line 409: RapA is not a repressor of kinA. RapA counteracts the effect of KinA.

R13: We fully agree. The word “it” in the sentence was meant to refer to the sporulation phosphorelay. We rephrased it as follows to avoid ambiguity: *Two central regulators of the delay-time distribution are the histidine kinase KinA and the RapA phosphatase, which activate and inhibit signaling, respectively.*

C14: Lines 466-474: Use of the word “memory” here isn’t explanatory. Ald is present at variable concentrations; it isn’t a question of whether the cell remembers that it has a certain concentration of Ald.

R14: We agree that more explanation of the term is required. We have now replaced the first two paragraphs by a focused discussion of spore memory that results from carry-over of proteins from progenitor cells into the spore.

C15: Line 520: The presence and activities of the different kinases are a response to different environmental conditions.

R15: We agree that the different kinases respond to different environmental conditions. However, this sentence is intended to make an evolutionary statement. Evolutionary theories suggest that duplication of genes often precedes and enables the evolution of novel functions mentioned by the referee. We believe that our model could help to explain how a gene duplication event can provide a survival advantage for sporulating bacteria (even before gene divergence has occurred). We have included references to recent work by Even-Tov and coworkers to provide context for this statement.

C16: Line 528: It isn’t clear what “more efficient spores” means. Perhaps the intent is to refer to cells that sporulate more efficiently.

R16: We agree that this requires further explanation. We have rephrased the sentence as follows: *We predict that this strain shows a decreased spore yield but generates spores that are capable of reviving more efficiently to weak nutrient stimuli, and we speculate that the gut environment may select for such spores.*

C17: Line 552: It isn’t clear what “quality” means. From the bacterium’s point of view, quality would refer to the level of resistance to heat, chemicals, etc.

R17: The referee points out that quality can have a variety of meanings in different contexts. We focus on the ability of a spore to grow out in response to nutrients, and use

this property as a measure for the quality of a spore, which our present experiments address. We certainly agree that the ability to resist environmental challenges during the dormancy period is another important aspect of the overall quality of a spore. This will be addressed in a future study.

C18: Line 580: Would addition of a 1/6 volume of CH or LB induce germination in a test tube?

R18: Yes.

C19: Line 591: Change “stop” to “slow down”.

R19: In the context of the mathematical model, we in fact assume that enzyme production ceases and that the enzymes made prior to this point are stable. This idealization greatly simplifies the model, and is not in conflict with our present data.

C20: Line 622: How much swelling was taken as an indication of the onset of germination?

R20: We have investigated both changes in swelling as well as changes in brightness. We measured the changes in spore width, which increases by about 40% during germination, while the brightness drops to about 56%. We refer to a spore as germinating, when its brightness has dropped below a threshold value of 70%. We updated the Methods section accordingly.

C21: Supplementary material: The movies are exceptional. But it is unclear what the actual conditions are. Is it correct that the medium for the agarose on which the cells were placed is SM medium with reduced glutamate? Why was L-alanine added to the medium? If the same cells were used to inoculate a liquid culture at the equivalent concentration in the same medium, how quickly would sporulation occur and what fraction the cells would sporulate?

R21: We thank the referee for responding so positively to our movies. The referee has described the medium composition correctly. During the initial stages of the project we refined the medium to optimize the time-lapse conditions. We found that reduction of the glutamate concentration helps to keep cells in a monolayer. The addition of L-alanine to the sporulation medium leads to a more pronounced revival phenotype. Following the suggestion of the referee, we have now mimicked the pad conditions in a liquid culture using the exact same medium and a corresponding inoculum of 6×10^5 cells/ml (5 ml medium in a test tube). We find that 10^8 spores/ml are generated under these conditions. This indicates that there is substantial replication before the onset of sporulation.

Reply to the comments of referee 2

C1: This ms. is directed at a central question of microbiology: how does the phenotypic diversity of single cells affect their future behavior. In specific, the ms. reports observations regarding the effect of sporulation timing on the ability of dormant spores to germinate. The study uses novel methodological innovations that allow for the first time an analysis of the germination behavior of single spores and their developmental history that provide new insights into the system. In addition, their use of modeling to make explicit predictions is impressive and will be a great example for how modeling is useful for understanding biological systems. The ms. is well-written and data convincing. I do however, have several issues that the authors should consider.

R1: We thank the referee for the assessment of our manuscript and the very constructive criticism, which has been of great help to us during the revision process. We address each point below.

C2: Why are the times for the various germinants different in Fig, 2D? Also, why do the authors think that there is only ~2-fold effect with LB, much different than the absolute effect of CH or L-ala? That is, why should different germinants act qualitatively differently?

R2: The reasons for choosing different time intervals to evaluate the revival frequency for different nutrient solutions are of a technical nature. In rich medium - such as LB - there is rapid cell division once spore outgrowth has occurred. In the vicinity of dividing cells, the remaining spores are then displaced and may even be obscured. Thus, with increasing time after the nutrient upshift, it becomes more and more difficult to confidently track spores that have not yet grown out. To counter this effect, for each medium used, we set the time interval to the maximum compatible with faithful tracking of the spores and acquisition of adequate statistics for each condition. We have now added a sentence to the Methods section to clarify this point.

The referee points out unexplained quantitative differences between the spore revival responses of early and late spores when using different nutrient solutions that are seen in Fig. 2D. The difference between early and late spores is on the order of 2-fold, while a much stronger effect is seen with L-alanine. It is in general very difficult to compare the different conditions, as both the type of nutrient and its concentration matter (see Fig. 3, Extended Data Figure 1). It is conceivable that a rich medium such as LB allows the spores to utilize several alternative pathways to generate the energy/key metabolites that are required for outgrowth, while spores that can utilize fewer substances, e.g. when stimulated with alanine, might be more constrained, thus giving rise to a more

pronounced phenotype. A more detailed study will be required to answer this question. Therefore, Fig. 2D aims to make a qualitative statement only: under all tested conditions the revival frequency of early spores was higher than that of late spores. In the revised manuscript we have streamlined the presentation of this data to emphasize this point.

C3: I find Fig 4A confusing – this distribution is not really “bi-modal” (line 228) – and what is the cutoff? How did they choose it?

R3: We agree that the fluorescence distribution from spores generated in liquid media shown in Fig. 4A is not clearly bimodal. However, it shows a pronounced shoulder, and this shape does suggest the presence of two subpopulations, although, owing to the shape of the distribution, it is difficult to define these in a clear-cut way. For the revised manuscript we have chosen the median of the fluorescence as a cut-off (see Fig. 5A (new numbering)). The qualitative results remain unchanged: spores that were generated early have a higher revival frequency than late spores.

C4: Why do the effects they see with the RapA reporter (low rapA vs. high rapA) on germination (Fig. 4B) not as strong as (Fig. 2B; early vs. late spores)

R4: The referee asks why the spores shown in Fig. 4B (now Fig. 5B) and the spores shown in Fig. 2B show different revival frequencies. Most probably the different culture conditions during sporulation and/or spore revival account for these effects. The spores in Fig. 2 were generated on an agarose pad composed of SM*. In contrast, for Fig. 5 the spores were generated in liquid SM media. Spores that are produced under these conditions show quantitatively different fluorescence distributions from the *rapA* reporter (Fig. 4C vs. Fig. 5A), indicating that the sporulation dynamics are sensitive to changes in environmental conditions. Moreover, owing to the different nature of the experiments, in the case of Fig. 2B spore revival was stimulated by adding nutrients to the “spent” SM* pad, while spores generated in liquid media were stimulated on fresh SM pads. We do not have any indication that the presence of the reporter influences the response of spores (see Fig. 4, Supplementary Video 2). Given that that the revival behavior of spores is known to depend on sporulation and upshift conditions, it is not entirely unexpected that the results should be quantitatively different. The important point is that we observe that early spores are more likely to revive than late ones, regardless of whether spores were produced on agarose pads or in liquid media.

C5: While the author’s talk quite a bit about a gfp fusion to ald promoter as a reporter for Ald levels (e.g., line 332), they do not present any data that there is a correlation between Ald

protein levels and reporter activity. In addition, they state that the decay in the reporter is reflective of a decrease in expression from the *ald* promoter (line 335) – but in the absence of any data regarding the stability of the reporter under these conditions, a direct correlation is only an assumption.

R5: The referee points out that a fluorescent promoter fusion may not report on Ald levels. We agree that direct proof for the correspondence between reporter and Ald protein was lacking. For the revised manuscript we have addressed this point with the help of an Ald-mCherry fusion protein that is expressed from its native promoter in the time-lapse experiment (Supplementary Video 6). This shows that the fluorescence trajectories from cells expressing the fusion protein are indeed comparable to those measured with the promoter fusion. In the revised manuscript the data on the *ald* promoter fusion (now Extended Data Figure 8) is used as the basis for the spore memory model in Fig. 6A. The new data on the fusion protein are summarized in Fig. 6B. We would also like to mention that the Ald protein is predicted to be stable, based on its amino acid sequence (<http://ca.expasy.org/tools/protparam.html>).

C6: The posit that Ald levels in “early” spores are higher than those of “late” spores (line 336-8). They could test this assumption by using FACS to isolate the separate populations and determining the relative Ald levels.

R6: We have taken up the referee’s suggestion that we should provide further evidence that early spores have higher Ald levels than late spores. To do so, we utilized an Ald-mCherry fusion and followed the fluorescence dynamics by time-lapse microscopy. The data is presented in Fig. 6B in the revised manuscript. We indeed find that early spores exhibit a higher fluorescence level than late spores. Moreover, we detected significantly more Ald-Cherry in spores derived from the accelerated sporulation mutant (using *kinA* overexpression) relative to spores generated from the slowed-down mutant (using *rapA* overexpression); see Fig. 7D in the revised manuscript.

C7: They should test that the *ald* induction (Fig. 6D) affects alanine-dependent germination specifically (e.g., try other germinants).

R7: The referee asks whether *ald* causes a specific defect in response to alanine. We observed no significant differences in spore revival frequency when we stimulated the spores with AGFK (Extended Data Figure 7). We would like to emphasize that under our conditions *ald* appears to specifically affect outgrowth. Induction of *ald* did not reverse the differences between early and late spores with respect to their germination kinetics in response to L-alanine (Extended Data Figure 6). For the revised manuscript, we also

analyzed the behavior of an *ald* knock-out strain. This confirmed that *ald* affects outgrowth (see Extended Data Figure 4 and Supplementary Video 4). Together, these data suggest quite strongly that Ald specifically controls alanine-induced outgrowth under our experimental conditions. To make this clearer we have introduced a paragraph describing the *ald* mutant data before introducing the data on memory.

C8: Although their modeling in Fig. 7 is very intriguing, it would be strengthened if they could provide more than a single point (Fig. 7D) for KinA and RapA induction; a full dose-response curve is not necessary but a second point for each would greatly increase confidence that they are on the right track. They could also try the Bs natural isolates (line 527) that apparently exhibit accelerated sporulation to see how they fit on the curve.

R8: The reviewer asks for further support of our modeling on the basis of additional data obtained by varying the induction levels of KinA and RapA. We followed this advice by taking advantage of the Ald-mCherry fusion protein, which was constructed as part of the revision. In this way we generated not only an additional data point for each case, but could also measure fluorescence to provide a proxy for the Ald levels in the spore. The data is presented in a revised Fig. 7d and further supports our model. We believe that the experiment regarding the natural isolate should be addressed in another manuscript.

C9: Line 149 “late” should be italicized. **R9: Corrected.**

C10: Line 512 “driving” should not be italicized. **R10: Corrected.**

Reply to the comments of referee 3

C1: In this manuscript the authors demonstrate that after an asynchronous sporulation, the spores that germinate relatively early are derived from cells that completed sporulation relatively early. This is an interesting observation. However, there are several significant difficulties. First, as detailed below, several experiments are compromised by a misunderstanding of the role of L-alanine, which does not support outgrowth. This has important implications for the authors' interpretation of some experiments, which are not convincing. There are other difficulties with experimental interpretation. Importantly, the manuscript is quite difficult to read in some places. This includes the use of informal terminology. Overall, some important interpretations and claims of the paper are not convincing.

R1: We thank the referee for the critical assessment of our manuscript and for pointing out where our manuscript requires improvement and appreciate the opportunity to clarify apparent misunderstandings.

We studied the relationship between sporulation timing and spore revival, which is completed upon successful outgrowth of the spores (Fig. 1, Fig. 2). Spore revival is traditionally partitioned into two stages: germination (Ca^{2+} -DPA release, rehydration, and cortex hydrolysis) and outgrowth (re-initiation of metabolism after germination is completed, escape from the spore coat, and elongation). While we do show that sporulation timing affects germination (Fig. 3A in the revised manuscript) as pointed out by the referee, we also report – and mainly focused on – effects on outgrowth. Moreover, we were able to provide insights into the mechanism (Fig. 3B – Fig. 6) underlying the latter effect and the consequences thereof (Fig. 7).

We agree that L-alanine is a trigger for germination. However, we disagree with the statement that alanine “does not support outgrowth”, if it is taken to mean that alanine *exclusively* serves as a germination trigger and cannot be utilized for outgrowth under any experimental conditions. We believe, on the other hand, that alanine *can* be utilized during outgrowth, although other factors that are present in the spent media might also be required to successfully complete outgrowth. Moreover, the referee seems to agree with this latter statement (see specific comment C18), which is also backed up by the previous literature.

Alanine is clearly an important (co)factor that contributes to outgrowth under our experimental conditions. Several lines of evidence reported in our first manuscript and

additional experiments conducted during its revision support this assertion. We summarize these findings below:

- First, if alanine were only a germination trigger and the spent medium alone was fueling outgrowth, then all spores should grow out after being exposed to alanine and germinating. However, upon addition of low levels of alanine, spores germinate but do not grow out (Fig. 3A). Moreover, a fraction of these spores proceed with outgrowth after receiving a *second* alanine stimulus (Fig. 3B). Finally, starved non-growing (non-sporulating) cells also resume growth upon the addition of alanine. These findings are consistent with reports in the literature, which state that *B. subtilis* can utilize alanine for outgrowth on spent media, and that alanine can support the growth of vegetative cells.
- Second, our fluorescent reporter experiments suggest that germinated spores take up alanine. In the presence of alanine, the *ald* promoter is activated. In spores that carry a fluorescent P_{ald} -Ald-mCherry reporter, fluorescence rises during spore revival when spores are induced with alanine. Thus, gene expression from the *ald* promoter suggests the uptake of alanine during outgrowth in our experiment. See Supplementary Video 6.
- Third, the enzyme Ald has been detected in spores previously. We also find that spores from a strain expressing an Ald-mCherry protein fluoresce - and that this correlates with their ability to grow out in response to alanine (Fig. 6c).
- Finally, the functions of alanine as a germinant and as a (co)factor for outgrowth can be genetically separated. Under our experimental conditions, *ald* mutant cells sporulate and the resulting spores germinate comparably to the *wt*. However, the germinated *ald* spores have an outgrowth phenotype (Supplementary Video 4, Extended Data 4). Moreover, inducing expression of Ald prior to sporulation increases the probability of successful spore outgrowth (Supplementary Video 5, Extended Data 4, Fig. 6d).

Thus our data shows that - under our experimental conditions - alanine indeed contributes to outgrowth. How alanine contributes to outgrowth in molecular terms is an interesting question that goes beyond the scope of our study. For two reasons, it seems likely that the germinated spores utilize alanine to fuel outgrowth. First, germinated spores contain Ald and the known biochemical function of Ald is to convert alanine to pyruvate; secondly, the germinated spores seem to take up alanine.

In the revised manuscript we have made an effort to more clearly communicate these points. Specifically, we explicitly define the relationship between revival, germination and outgrowth frequency, and now discuss our results relating to the effects of alanine stimulation of spore germination and outgrowth earlier in the manuscript (Fig. 5 -> Fig. 3). In addition, we have included a section in which we describe the phenotype of *ald* knock-out and overexpression mutants (Extended Data Figure 4,5,6,7) before introducing the data that support our contention that Ald acts via phenotypic memory. We hope that these changes make our arguments clearer and help the reader to focus on our main findings, which concern the role of spore memory in coupling sporulation to spore revival.

C2: There is no acknowledgement of extensive past work on variation in sporulation and germination (particularly from the Setlow lab) is made. This is important contextual information and it is surprising that this work is not discussed.

R2: The referee points out that work by Peter Setlow on “variation in sporulation” and germination would be required to provide more context for the current study.

The phrase “*variation in sporulation*” can refer either to environmental conditions or to variability in sporulation timing. Work by Setlow and others demonstrates that the environmental conditions during sporulation affect the properties of spores. In the Introduction, we refer the reader to an excellent review by Hornstra et al. that summarizes much of this work. Moreover, this article also proposed the hypothesis that variability in sporulation timing could influence the properties of the resulting spores as a result of changing environmental conditions. However, to the best of our knowledge, there is no experimental evidence for this hypothesis. We would like to point out here that our study does not provide any evidence for this hypothesis either. While our data does show that variation in sporulation timing affects spore revival, altered sporulation conditions are *not* the major factor that account for the striking difference in the spore revival frequencies of early and late spores in response to L-alanine.

Instead, we identify Ald as a component of the spore’s proteome that is carried over from a spore’s progenitor cell and this effect gives rise to an intrinsic spore memory that provides the coupling between sporulation timing and spore revival. We would also like to point out that most of our mechanistic findings are concerned with effects on spore outgrowth, not germination.

We made several changes to the Introduction, the Results section and the Discussion to improve our presentation. Specifically, as requested by the referee, we include an additional reference to more recent work by the Setlow lab which shows that environmental conditions affect the properties of spores. Moreover, we now introduce the idea that carry-over effects from progenitor cells could result in history-dependent effects in the Introduction by referring to proteomic studies (Kuwana et al. 2002, Liu et al. 2004, Bergman et al. 2006), which found that a considerable portion of a spore's proteome is not expressed during sporulation but is instead derived from progenitor cells. We hope that these changes help to dispel misunderstandings and facilitate placing of our work in its proper context.

Reply to the specific comments from referee 3

C3: The abstract is very difficult to understand. This is because of the use of obscure terminology and phrases. Examples of obscure terminology include “Phenotypic memory”, “imaging life histories” and “history-labeling” (26-27), which are only defined later on. Obscure phrases include “sporulation timing”, “kinetically favored”, “early spores” and “late spores” (27-28). More importantly, the abstract does not provide any experimental data or specifics and, as a result, does not do an adequate job of conveying the real nature of the results obtained, or the empirical basis of any of the claims.

R3: The referee is concerned that obscure terminology and phrasing compromise the readability of the abstract and asks for more experimental data in the abstract. To communicate our findings concisely, we made use of terms from different fields. The term “phenotypic memory” was introduced by Jablonka et al. in 1995 and has been used in numerous publications since then, including recent work on bacteria, to describe history-dependent effects in cells. “Sporulation timing” and the term “early spores” have also been used previously by others in the sporulation community. To improve clarity in the revised abstract, we tried to either define or avoid some of these terms altogether. We have also made an effort to improve the description of our results in line with the guidelines of the journal, which ask for a non-technical summary of results.

C4: 50: “phenotype” is the wrong word.

R4: Thanks, we replaced “phenotype” by “cell type”.

C5: 59: what does “state transition” refer to?

R5: Thanks, we rephrased this sentence using the term “conversion from one cell type to the other”.

C6: 61-62: it is essential here to avoid making an unsupported claim; bet-hedging per se has never been demonstrated. Rather, the idea that these behaviors are a type of bet-hedging (and, therefore, adaptive) remains a speculation (although an intriguing one).

R6: The referee points out that bet-hedging has yet to be demonstrated experimentally. We are aware of this and agree. That is why we wrote: “cell-to-cell heterogeneity in the propensity to undergo *conversions from one cell type to the other* **has been proposed to serve adaptive bet-hedging.”**

C7: 73: What does heterochronous mean? Asynchronous? If so, use that more conventional word.

R7: Heterochronous means “occurring at different times or stages”. The term “heterochronous” is widely used by the sporulation community (see Refs. Chastanet et al., de Jong et al.). It relates to a common concept from developmental biology. We thus prefer to use “heterochronous” here.

C8: 87-88: The text should not give the implication that this sort of microscopy is novel; numerous studies have visualized sporulation and germination dynamics in single cells.

R8: We fully agree that both processes have been visualized before. However, to the best of our knowledge, this has never been done on the same cells. Indeed we developed a correlative imaging approach for this study in order to track individual spores from sporulation all the way to spore outgrowth. We have rephrased this paragraph to clarify this point.

C9: 129: This is an important point; it is well established that L-alanine is not at all a nutrient. This matters to experimental interpretation. By “revival response”, do the authors mean “outgrowth”? Here and elsewhere, switching between terms in this way and, especially, the use of informal terminology, leaves reader confused not only as to the specific assertions being made, but also regarding the overall logic.

R9: We thank the referee for drawing our attention to the fact that we had not included a definition of spore revival in the main text. Since the term ‘revival’ is sometimes used informally in the literature to refer to germination also, we agree that this could be confusing. However, we consistently use ‘revival’ to refer to the overall process of turning a dormant spore into a vegetative cell. Thus, successful outgrowth of the spore falls within this definition. However, since outgrowth refers to a specific stage of revival that sets in *after* germination, we prefer to use the term “revival response” to set it apart from the “outgrowth response”. In a limited set of experiments, we investigated the

effects on revival and analyzed effects on germination and outgrowth (Fig. 3 in the revised manuscript). To improve clarity, we now define the terms *revival frequency*, *germination frequency* and *outgrowth frequency* at appropriate positions in the main text and include more details in the Methods section. Moreover, we introduce the data on germination and outgrowth in response to L-alanine earlier in the manuscript (Fig. 5  Fig. 3). As pointed out in R1, we disagree with the assertion that alanine cannot serve as a nutrient and we are not aware of any study that demonstrates the contrary.

C10: At an intuitive level, I understand the reason for the choice of the time point used to distinguish between early and late spores. However, this choice is not explicitly justified at this point, and it's not really clear that the conclusions of the paper are insensitive to the choice. What are the results of a sensitivity analysis to address this question? Specifically, how would this analysis affect figure 2D?

R10: We agree that partitioning of spores into early and late involves an arbitrary choice, and in principle, one could apply alternative partitioning schemes. For instance, spores may be split into an *early* and a *late* fraction at the median sporulation time. However, splitting the population using the median does not affect the quantitative results for Fig. 2D strongly as ~50% of all spores are already classified as early and late respectively using our heuristic criterion. Since our present choice of partitioning is then later justified by the fluorescence reporter, we prefer to retain the heuristic grouping criterion. In response to a similar suggestion from referee 2 we applied the median and re-analyzed the data for spores that were generated in shake-flask culture (Fig. 5B). As expected, this alternative partitioning of spores changed the results quantitatively, but not qualitatively. Under all conditions tested, earlier sporulation results in a higher revival frequency.

C11: 153: How is revival frequency defined?

R11: The revival frequency is defined as the fraction of all outgrowing spores $f_r = N_o/N_d$, where N_o is the number of outgrowing spores and N_d is the total number of dormant spores. The revival frequency is given by the product of the germination frequency $f_g = N_g/N_d$ (the fraction of dormant spores that germinate) and the outgrowth frequency $f_o = N_o/N_g$ (the fraction of germinated spores that grow out), where N_g is the number of *germinated* spores. To improve clarity we have revised the Materials and Methods section and also provide a definition in the main text.

C12: 176-177: I don't see how heat activation is relevant to the point being made.

R12: Thanks. This reference was omitted in the revised manuscript.

C13: 180: This does appear to provide good support for the earlier time choice. However, the data in figure 3C showing the fluorescence changes at about 20' are from a limited data set, and it's not entirely clear what we are looking at. Is this the combined data from two experiments, or two separate experiments that are, somehow, averaged? I think this matters, as the apparent break point within the data could be a fluctuation. If it is real, this would be interesting, but needs a better statistical validation.

R13: As requested, we have increased the sample sizes to improve the statistics and analyzed the fluorescence distribution from more than 900 spores. This confirmed that the fluorescence distribution is bimodal and that fluorescence correlates well with sporulation timing.

C14: 261 and following: I don't see how the experiments immediately prior to this conclusion strongly support the conclusion. I think the conclusion is a reasonable working model, and a better likely explanation for the phenomenon, but this is largely because it is already clear from earlier literature that sporulation results in variation in the spore population. But I don't see how alternative explanations are specifically excluded.

R14: The referee points out a lack of logical stringency when arriving at line 261. This statement was intended to refer to an entire set of experiments, not just to the experiments described under this sub-heading. In the revised manuscript we have omitted all subheadings to improve clarity and save space; we hope this has helped to clarify the context of the statement.

C15: 277: How was swelling measured? It appears that refractility was measured, not swelling.

R15: Given the time resolution of our experiments, there is no great difference between these two criteria. We have investigated both, changes in swelling as well as changes in brightness. We measured the changes in spore width, which rises by about 40% during germination, while the brightness drops to about 56%. We have updated the Materials and Methods section accordingly to provide more details on how germination was assessed.

C16: 279: It is imprecise to say that germination was faster. It appears that the completion of the conversion to the phase bright state was faster.

R16: Presumably the referee wants to point out that, in addition to changes in the optical properties, other changes required for successful germination occur in the spore that we do not directly observe. We fully agree with this assertion. However, it is quite common to refer to spores that have changed their refractive properties and have swelled as being germinated (see Ramirez-Peralta et al. 2012). We now include a precise definition of

germination frequency in the Materials and Methods section and hope that this is sufficient to place this statement in its correct context.

C17: 301 and following: I am unclear on the precise experimental question being asked. But, if the experiment hinges on L-alanine stimulating outgrowth, then the experiment is flawed, since L-alanine does not have this effect.

R17: The referee asks for clarification of the experimental assay in line 301 (described in Fig. 3B in the revised manuscript). We investigated whether spore history might affect spore outgrowth independently of its effect on germination. To this end, we focused on the outgrowth of *germinated* spores by measuring the outgrowth probability f_o (see R11). Spores were triggered for germination by a weak alanine stimulus and then ‘stalled’ in the germinated state. Subsequently, we applied a second alanine stimulus and measured the outgrowth frequency among the *pre-germinated* spores. Although early and late spores had the same final germination frequency, their outgrowth frequencies were distinct. We have revised the presentation of this section to clarify this point. As noted before, we disagree with the referee’s contention that the experimental design is flawed and refer the referee to R1 for our rebuttal on the effect of alanine on outgrowth in general.

C18: 324 and following: I don’t think this experiment demonstrates what is claimed, for the reason already cited and also because the presumption on line 335 is not compelling. Certainly, metabolism may depend to some degree on alanine. However, alanine alone does not support outgrowth, and the involvement of the spent medium in the effects seen here prevent a clear interpretation.

R18: The referee points out that spores should be unable to use L-alanine as the sole source of nutrients for spore revival. We agree that spores may require additional factors for successful outgrowth, which may be derived from the spent medium or lysing (mother) cells. We neither claim that alanine is able to support outgrowth on its own, nor do we aim to investigate the effect of alanine on outgrowth. A separate study will be required to clarify the exact mechanism by which L-alanine contributes to outgrowth. Please refer to R1 for more details.

The referee is furthermore concerned that the presented experiments using a fluorescent fusion to the *ald* promoter are not sufficient to justify the inference that sporulation timing affects the Ald levels in the spore. We agree that additional data on the protein level is required (see also the comments C4 & C5 by referee 2). Therefore, for the revised manuscript, we addressed this point with the help of a mCherry-Ald fusion protein that is

expressed from the *ald* promoter. This experiment first confirms that cells downregulate the expression of Ald (Fig. 6b, left). Line 335 offered an ‘explanation’ for this effect based on the known regulation of the Ald promoter: Since alanine activates the *ald* promoter via AdeR and our sporulation medium contains alanine, the *ald* promoter should be active initially, and as cells grow and deplete alanine from the medium, AdeR would be deactivated and the promoter would be turned down. More importantly, our new experiments with the Ald-mCherry fusion protein provide strong evidence that Ald is carried over from the progenitor cells into the spore (Supplementary Video 6). In particular, early spores have much higher mCherry fluorescence than late spores (Fig. 6b, center & right). Moreover, we detected significantly more Ald-Cherry in spores derived from a mutant in which sporulation is accelerated (owing to *kinA* overexpression) than in spores generated by a sporulation mutant in which the whole process is slowed down (using *rapA* overexpression), see Fig. 7D in the revised manuscript. Together, this data demonstrates a tight relationship between sporulation timing and Ald levels in the resulting spores.

C19: 342: Given the potentially pleiotropic effects of Ald, it is hard to see how one would uniquely interpret these data.

R19: The referee worries that *ald* could influence several traits simultaneously. We agree that this is a valid concern, as previous studies have found that the phenotype of an *ald* knockout under some conditions impairs sporulation, while under others spores form but exhibit a revival phenotype. Under our experimental conditions, overexpression of Ald affects alanine-induced outgrowth (Extended Data Figure 4), but has no impact on germination (Extended Data Figure 5) or the spore revival frequency in response to AGFK (Extended Data Figure 6). Moreover, for the revision, we constructed an *ald* knockout and analyzed the behavior of the *wt* and the mutant strain in a co-culture experiment (Supplementary Video 4). We found that, under our experimental conditions, *ald* cells sporulate well. Moreover, the mutant spores are also able to germinate. Strikingly however, unlike the *wt* spores, the mutant spores fail to grow out in response to L-alanine (Extended Data Figure 4). Together, this data provides strong evidence that – under our conditions – the effect of *ald* is very specific.

C20: 348-350: This is very confusing. By differentiation, do the authors mean sporulation? If so, is this a post-hoc analysis, looking retrospectively at cells that ultimately became what were shown to be early spores? I am unable to follow the description of the experiment.

R20: The referee asks us to clarify the Ald induction experiments. These experiments examine whether one can rescue alanine-dependent outgrowth by boosting Ald levels in the progenitor cells of late spores. Perhaps the best explanation is actually given by watching the Supplementary Video 5. We have now included an additional cartoon that illustrates the experiment (Extended Data Figure 5) and have substantially reworked the Results section. Specifically, we have split the presentation of the experiments into two parts. We discuss the first experiment (now presented in Extended Data Figure 5) to support the conclusion that *ald* causes a specific defect in the outgrowth defect of late spores in response to L-alanine and to point out that the coupling between sporulation timing and spore revival could be reversed in principle. We use the second experiment to support the conclusion that Ald contributes to a spore's 'memory' by demonstrating that the history of *ald* gene expression in progenitor cells affects the revival properties of spores (Fig. 6C). We hope that these changes to the presentation enhance clarity.

C21: 395 and following: I am not sure how useful the model really is. It is a dynamical description of the authors' interpretation of the data, but I don't think that this representation of the dynamics actually provides any support for any conclusion.

R21: The referee has doubts about the usefulness of the mathematical model. We agree that our mathematical model is a dynamical description of our hypotheses about the mechanism underlying the experimental observations. It goes beyond a mere description in words in that it allows us to explore the implications of our hypotheses quantitatively. For us, the mathematical model was particularly useful to investigate the consequences of memory on the coupling between sporulation and spore revival. The model predicted the emergence of a quantity-quality tradeoff (Fig. 7A). We then tested this prediction experimentally and found good agreement (Fig 7B,C). In the revised manuscript we provide further direct evidence in favor of the model by perturbing sporulation timing and measuring the numbers of spores, and the Ald levels in the spores, which relates to their quality, affecting their revival frequency (Fig. 7D). The combination of modeling and experiments provides the basis for our conclusion.

C22: 404 and following: It is difficult to be convinced by the interpretation put forward by the authors. The effects of the mutations documented here could have many interpretations, and don't seem to provide support for a unique interpretation.

R22: The referee is concerned that our set of mutant experiments do not allow for a unique interpretation. To provide further evidence in favor of our model, we conducted a set of additional experiments, which are briefly summarized below.

- **First, to test whether there is a relationship between spore yield and spore quality we chose two genes (*kinA* and *rapA*), which both affect sporulation timing, but whose proteins have different biochemical functions. For each gene, we varied the induction strength (following the suggestion of referee 2) to modulate sporulation timing more gradually. As predicted, spore yield and (alanine-dependent) spore outgrowth are anti-correlated, see Fig. 7D, top.**
- **Second, with the help of the Ald-mCherry protein we then quantified the average fluorescence of the mutant spores in comparison to the *wt*. As predicted by the model, the average fluorescence increases (decreases) as sporulation is accelerated (delayed), see Fig. 7D, bottom.**
- **Third, we excluded the possibility that the IPTG-based induction of *kinA* *per se* results in more responsive spores, see Supplementary Video 9. To this end, we induced *kinA* expression in the subpopulation of cells that in which sporulation was delayed. If *kinA* affects the outgrowth probability directly, the resulting spores should be able to grow out. However, in line with our model's prediction, we see that these spores have lower levels of Ald-mCherry and none of the late (*kinA*-induced) spores grew out.**

Together with our results showing that Ald plays an important role in establishing a phenotypic spore memory, these data provide strong evidence that sporulation timing affects the “quality” of the resulting spores, specifically their ability to grow out by utilizing alanine.

Reviewer #1 (Remarks to the Author):

The authors have responded very well to the previous criticisms and requests for clarity, but there remain some issues that need to be addressed. In many cases, it is for the purpose of making the experiments fully understandable by the reader.

1) The legends of the figures lack enough detail for the reader to fully understand the experiment in question. For instance, for the fusions, it is critical to state clearly in the figure legend whether a given fusion is carried in *B. subtilis* on a self-replicating plasmid or integrated into the chromosome. (The information in the Supplementary Material would require a reader to dig up other papers to know the answers.) Moreover, for integrated fusions, are they at unrelated sites (e.g., *amyE*, *sacA*) or at the site of the gene that provides the promoter? In addition, there appear to be differences between the fusions. That is, in one case only the promoter appears to be fused to the gene of interest whereas in another case the promoter and entire coding sequence are fused to the target gene. Finally, are there cases where there is more than one copy in the chromosome of the gene being investigated (e.g., *aldA*)?

2) What is the composition of the medium used for germination as in Fig. 6?

3) "Late" spores presumably sporulated more slowly because their intracellular pools of nutrients were higher than those of the faster sporulators. When they finally consume those nutrients and form spores, they may not germinate well because they have not been given enough time to form mature spores.

4) At least in some cases, "late" spores are kept in PBS for long periods of time. Since these spores may not have completed the spore formation process at the time of harvesting, they may not be able to do so without any organic molecules in the environment.

5) In Fig. 7D, the use of mCherry stability as a measure of Ald stability depends on the assumption that the two proteins have very similar stability. Is there any basis for this assumption?

6) On page 10, line 293: The word "not" should be deleted.

7) Fig. 3 legend, line 680: It would be clearer for the reader to reword as: "Germination frequency, defined as loss of refractivity, of early "

8) Page 2, line 22: The first word of the Abstract should be "Some".

Reviewer #2 (Remarks to the Author):

The authors have satisfactorily responded to my original critique so I strongly advocate for acceptance of the revised manuscript.

RESPONSE TO REVIEWER 1:

The authors have responded very well to the previous criticisms and requests for clarity, but there remain some issues that need to be addressed. In many cases, it is for the purpose of making the experiments fully understandable by the reader.

We thank reviewer 1 for the careful review of our revised manuscript and have followed the suggestions as detailed below.

1) The legends of the figures lack enough detail for the reader to fully understand the experiment in question. For instance, for the fusions, it is critical to state clearly in the figure legend whether a given fusion is carried in *B. subtilis* on a self-replicating plasmid or integrated into the chromosome. (The information in the Supplementary Material would require a reader to dig up other papers to know the answers.) Moreover, for integrated fusions, are they at unrelated sites (e.g., *amyE*, *sacA*) or at the site of the gene that provides the promoter? In addition, there appear to be differences between the fusions. That is, in one case only the promoter appears to be fused to the gene of interest whereas in another case the promoter and entire coding sequence are fused to the target gene. Finally, are there cases where there is more than one copy in the chromosome of the gene being investigated (e.g., *aldA*)?

R1: We agree that these details are important. All of our fluorescent reporter constructs were integrated at an ectopic locus (*amyE*, *sacA*, *ppsB*) and are present in single copy in the chromosome. All fluorescent reporters are promoter fusions with the exception of the experiments involving Ald-mCherry (Fig. 8 and Fig.10). These experiments seek to provide evidence that the Ald protein is indeed carried-over from the progenitor cells into the spore. The fluorescently-tagged Ald is expressed from its own promoter and the construct was integrated into the *amyE* locus (Fig. 8) or the *sacA* locus (Fig. 10D), respectively. The native Ald is thus present in these strains as well. To improve clarity we have revised all legends and also included the strain number in each case.

We furthermore included more details on the reporter construction in the materials and methods section. This section has been expanded to include the text previously included in the Supplementary Material. Relevant genotype information on all vectors, plasmids and strains can be found in the Supplementary Tables 1-3. These tables have been restructured to separate information on integration vectors (Supplementary Table 1) from derived plasmids (Supplementary Table 2) and strains (Supplementary Table 3).

2) What is the composition of the medium used for germination as in Fig. 6?

R2: We used L-alanine to induce the upshift. We updated the legend accordingly.

3) "Late" spores presumably sporulated more slowly because their intracellular pools of nutrients were higher than those of the faster sporulators. When they finally consume those nutrients and form spores, they may not germinate well because they have not been given enough time to form mature spores.

R3: Most late spores had more than 24 hours of time to mature post-release from the mother cell in our time-lapse experiment before the upshift (e.g. Supplementary movie 1). At least with respect to Ald-dependent outgrowth in response to L-alanine the differences cannot be explained by failed maturation as we can reprogram late spores to revive better than early spores (Fig. 6C). However, we agree that that post-mother cell release maturation remains a possibility that could explain the different germination kinetics between early and late spores. Future experiments are required to investigate this point.

4) At least in some cases, "late" spores are kept in PBS for long periods of time. Since these spores may not have completed the spore formation process at the time of harvesting, they may not be able to do so without any organic molecules in the environment.

R4: This experiment refers to spores that were produced in a liquid shake-flask culture and harvested after 4 days. Sporulation within the whole culture was completed after approximately 48 hours. Thus, even the resulting late spores would have had at least 48 additional hours prior to harvest (and transfer to PBS) to proceed with putative additional maturation. Moreover, if spores are kept in spent sporulation medium for extended periods of time, some spores may be able to germinate, which could then bias the population structure. Hence we think that incubating the spores in sporulation medium for 4 days is a good compromise between preventing unwanted germination and still providing sufficient time for possible post-mother cell lysis maturation. However, we agree that we cannot exclude this possibility entirely. Future experiments are required to address this point.

5) In Fig. 7D, the use of mCherry stability as a measure of Ald stability depends on the assumption that the two proteins have very similar stability. Is there any basis for this assumption?

R5: In principle a fluorescent tag could alter the properties of a protein, including its stability. We can thus not strictly exclude that the Ald-mCherry protein will be more or less stable than the native Ald-protein. However, protein stability predictions with ProtParam tool of the ExPASy server (<http://ca.expasy.org/tools/protparam.html>) classify both proteins as stable with a stability index of 24.96 and 27.99 for native Ald and the Ald-mCherry fusion, respectively. Thus fluorescence of Ald-mCherry should correlate to the actual amount of Ald in the cell.

6) On page 10, line 293: The word "not" should be deleted.

R6: Thanks. We removed “not” from this sentence.

7) Fig. 3 legend, line 680: It would be clearer for the reader to reword as: "Germination frequency, defined as loss of refractivity, of early "

R7: We agree and have revised the caption as requested.

8) Page 2, line 22: The first word of the Abstract should be "Some".

R8: We agree and have revised the abstract as requested.